# Permafrost degradation at two monitored palsa mires in north-west Finland

Mariana Verdonen[1], Alexander Störmer[2,4], Eliisa Lotsari[1,3], Pasi Korpelainen[1,4], Benjamin Burkhard[2,4], Alfred Colpaert[1], and Timo Kumpula[1,4]

[1]Department of Geographical and Historical Studies, University of Eastern Finland, Joensuu, 80101, Finland
[2]Institute of Physical Geography and Landscape Ecology, Leibniz University Hannover, Hannover, 30167, Germany
[3]Water and Environmental Engineering, Department of Built Environment, Aalto University, Aalto, 00076, Finland.
[4]Kilpisjärvi Biological Station, University of Helsinki, Kilpisjärvi, 99490, Finland.

**Correspondence:** Mariana Verdonen (mariana.verdonen@uef.fi)

**Abstract.** Palsas and peat plateaus are expected to disappear from many regions, including Finnish Lapland. However, detailed long-term monitoring data of the degradation process on palsas are scarce. Here, we present the results of the aerial photography time series analysis (1959–2021), annual RTK-GNSS and active layer monitoring (2007–2021), and annual Unoccupied Aerial System surveys (2016–2021) at two palsa sites (Peera and Laassaniemi, 68° N) located in north-west Finland. We analysed temporal trends of palsa degradation and their relation to climate using linear regression. At both sites, the decrease in palsa area by -77 % to -90 % since 1959 and height by -16 % to -49 % since 2007 indicate substantial permafrost degradation throughout the study periods. The area loss rates are mainly connected to winter air temperature changes at Peera and winter precipitation changes at Laassaniemi. The active layer thickness (ALT) has varied annually between 2007 and 2021 with no significant trend and is related mainly to the number of very warm days during summer, autumn rainfall of previous year, and snow depths at Peera. At Laassaniemi, the ALT is weakly related to climate and has been decreasing in the middle part of the palsa during the past eight years despite the continuous decrease in palsa volume. Our findings imply that the ALT in the inner parts of palsas do not necessarily reflect the overall permafrost conditions and underline the importance of surface position monitoring alongside the active layer measurements. The results also showed a negative relationship between the ALT and snow cover onset, indicating the complexity of climate–permafrost feedbacks in palsa mires.

## 1 Introduction

Permafrost degradation manifested by widespread thermokarst activity (Kokelj and Jorgenson, 2013), warming ground temperatures (Biskaborn et al., 2019) and (mostly) increasing active layer thicknesses (ALT) (Smith et al., 2022) is taking place throughout its extent in the polar regions, mountain environments and central Asia. Consequently, an extreme loss of permafrost landscape features is projected because of climate warming and the increase in precipitation in the northern regions (Aalto et al., 2014; Karjalainen et al., 2020; Fewster et al., 2022; Könönen et al., 2022). Ecosystem-protected permafrost

features located at the margins or even outside their climatic envelopes, such as palsas and peat plateaus, are particularly sensitive to climatic changes (Luoto et al., 2004a; Shur and Jorgenson, 2007; Meier, 2015).

Both peat plateaus and palsas are characterized by a thick, upheaved layer of peat and perennially frozen core (Zoltai and Tarnocai, 1975; Meier, 2015). The main differences between peat plateaus and palsas are in their topographical profiles. Peat
plateaus are usually extensive, flat-topped and less than three metres high, while peat mounds with a smaller areal extent but a more dome-shaped profile are deemed palsas. In the Fennoscandian context, both are often called palsas due to the similarity of their formation processes and common occurrence within the same peatland areas (Seppälä, 2011; Meier, 2015). Additionally, smaller mounds identified as separate palsas can also be disintegrated parts of larger peat plateaus (Borge et al., 2017).

Palsa mires are important features of subarctic landscapes. The heterogeneity of habitats makes them particularly biodiverse ecosystems (Luoto et al., 2004b). Degradation of palsas and peat plateaus alters mire topography, vegetation structure, hydrological processes and biogeochemical cycles (Christensen et al., 2004; Turetsky et al., 2007). Loss of dry, elevated surfaces also diminishes palsa mires' cultural and provisioning ecosystem services, such as wild berry picking and reindeer pastures (Way et al., 2018; Näkkäläjärvi et al., 2020).

The rapid lateral degradation of palsas and peat plateaus during the past several decades has been reported from Fennoscandia (Zuidhoff and Kolstrup, 2000; Meier, 2015; Borge et al., 2017; Olvmo et al., 2020) and North America (Payette et al., 2004; Jones et al., 2016; Mamet et al., 2017). The occurrence of widespread permafrost degradation also before the time covered by aerial photography time series is indicated by the presence of thermokarst ponds in the earliest available aerial photographs (Luoto and Seppälä, 2003; Payette et al., 2004; Borge et al., 2017).

The periods of most rapid lateral degradation vary depending on the studied region, methodology and temporal coverage of aerial photography time series used to map the extent of palsas and peat plateaus. A comparison of palsa degradation rates (area loss in % a$^{-1}$) during different periods and respective climatic conditions has linked rapid lateral degradation to increases in precipitation and temperature (Payette et al., 2004; Olvmo et al., 2020). On the contrary, Borge et al. (2017) did not find a correlation between the degradation rates and climatic parameters.

A delineation of the palsa extent from orthophotos does not provide information about the height and volume of palsas unless stereophotographs are available. High-resolution Light Detection and Ranging (LiDAR)–based Digital Elevation Models (DEMs) are therefore a great asset for (regional) palsa inventories (Wramner et al., 2015; Ruuhijärvi et al., 2022). The increasing availability of Unoccupied Aerial Systems (UAS) allows even more frequent and detailed mapping and change detection of both the extent and topography of palsas and peat plateaus (Martin et al., 2021; Fraser et al., 2022).

Active layer thickness (annually thawing and freezing ground layer underlain by permafrost) is one of the main parameters commonly used to monitor permafrost conditions (Brown et al., 2000). In the Northern Hemisphere, the ALT varies from a few decimetres in the High Arctic to several metres in the mountainous regions at mid-latitudes (Luo et al., 2016). Analyses of the long-term trends showed an increase in the ALT, especially in the Qinghai-Tibetan Plateau (Luo et al., 2016), Greenland and Scandinavia (Strand et al., 2021), West Siberia (Smith et al., 2021) and the interior of Alaska (Nyland et al., 2021). The lack of a significant change in the ALT on the Alaskan North Slope and the Mackenzie River delta has been (partially) attributed to the effect of thaw consolidation following the melting of ice in the transient layer at the boundary between the active layer and the permafrost table (Shiklomanov et al., 2013; Streletsky et al., 2017; O'Neill et al., 2019). Some active layer data are also available from palsas and/or peat plateaus (Åkerman and Johansson, 2008; Sannel et al., 2016; Mamet et al., 2017). Based on these studies, summer air temperatures and thaw degree days, and in some cases freezing degree days and snow accumulation, seem to be the most important meteorological predictors of the ALT. More than decade-long ALT records from palsas are scarce, however.

In this research, we present the results of long-term monitoring of two palsa mires in the Kilpisjärvi area, north-west Finland inspected annually in late summer. The aim of our study is to investigate permafrost degradation dynamics and their relation to climatic parameters. In particular, we aim to answer the following questions: (1) How did the lateral extent of palsas (1959–2021), palsas' height (2007–2021) and the active layer thickness (2007–2021) change during the investigation periods? (2) Can the detected changes be related to climatic drivers and if yes, which?

The study consists of three overlapping periods of decreasing length and increasing data availability. Aerial photographs with various intervals are available since 1959 and daily meteorological observations are available since 1960. The annual active layer monitoring grids including palsa surface measurements with high accuracy Real-Time Kinematic (RTK) GNSS were established on two palsas in 2007. For the last six years (2016–2021), the UAS RGB data have enabled capturing changes in palsas at ultra-high (< 5 cm) spatial resolution. This unique data set that includes multiple spatial and temporal scales allows comprehensive analysis of both lateral permafrost degradation and active layer dynamics of palsas in Finland.

## 2 Study sites

The study sites are located in the Kilpisjärvi area, the northwestern part of Finnish Lapland (Fig. 1a). The mean annual air temperature measured at the Kilpisjärvi weather station is -1.38 °C (1990–2020; FMI, 2022). The average air temperatures of the coldest (January) and warmest (July) months are -12.27 °C and 11.66 °C, respectively. The mean annual precipitation is about 514 mm, of which around 240 mm falls as snow (FMI, 2022).

Kilpisjärvi is located on the pre-alpine belt of the Scandinavian Mountains. The general landscape is characterized by flat summits typical for old peneplain surfaces and valleys with mires that have usually less than a 2 m thick peat layer (King and

Seppälä, 1987). This topography supports the formation of cold air ponds in the valleys and, in combination with low snow depths, favours the development of palsa mounds (King and Seppälä, 1987). South–southeast is the prevailing wind direction during the wintertime (November–April) as measured at the Kilpisjärvi weather station (FMI, 2022).

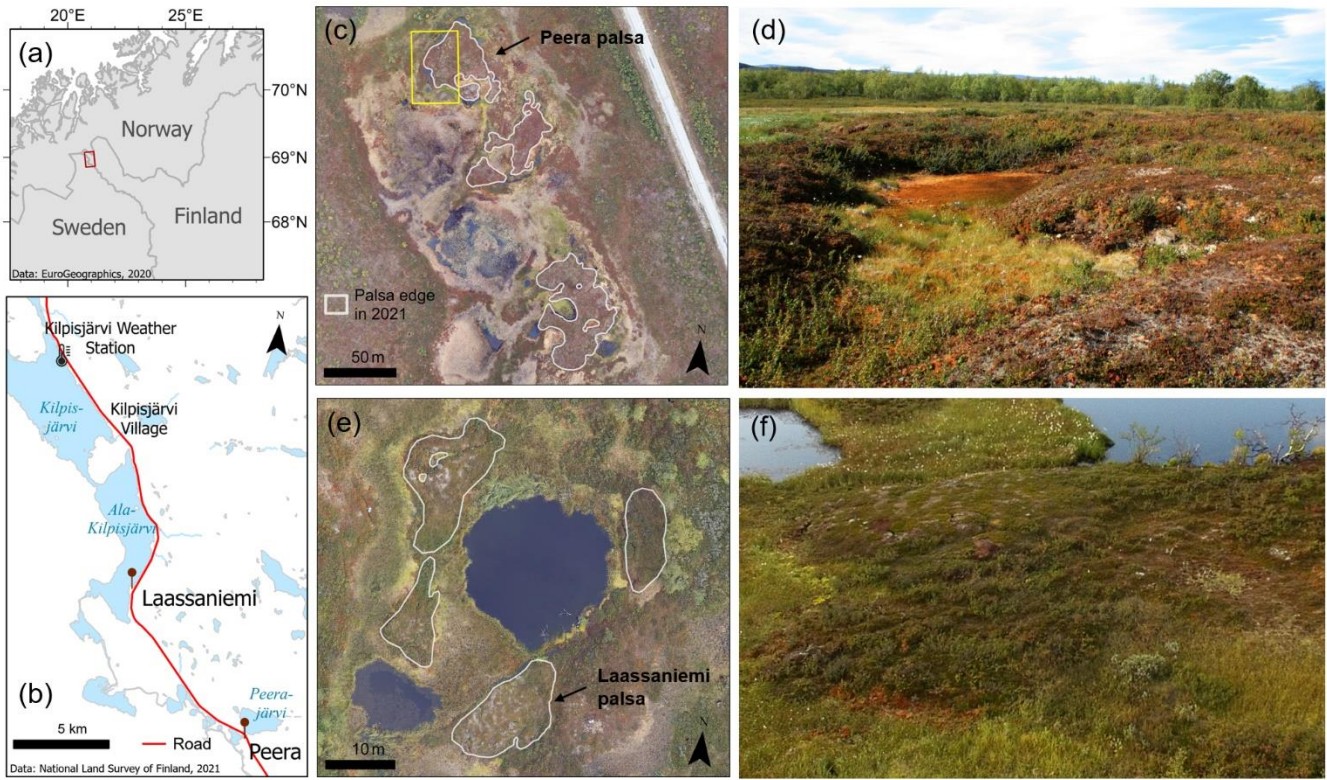

**Figure 1.** (**a**) Location of the study sites in north-west Finland. (**b**) Climate data used in this study is from the Kilpisjärvi weather station, from which the distance to Laassaniemi and Peera is around 12 km and 20 km, respectively. (**c**) In 2021, there were four palsas in the studied part of the Peera palsa mire. The yellow rectangle indicates the area of the RTK-GNSS and active layer measurements on the Peera palsa. (**d**) The Peera palsa is around two metres high. Dry palsa surface is vegetated by lichens, dry mosses and dwarf shrubs, while the thermokarst depression in the middle of the photo is fully vegetated by sphagnum mosses. (Photo by M. Verdonen, August 2012). (**e**) Four palsas in the Laassaniemi palsa mire. The annual monitoring is focused on the southernmost Laassaniemi palsa. (**f**) Laassaniemi palsa is less than a metre high. Low profile and surrounding vegetation – dwarf birch shrubs in particular – complicate delineation of palsa's edges from aerial data and digital elevation models. (UAS photo by P. Korpelainen, August 2018).

## 2.1 Peera

The Peera palsa mire is located south of Peerajärvi Lake, at 68°52'41" N, 21°04'41" E and around 460 m a.s.l. The studied mire area is ca. 5.2 ha and has four palsa mounds, which used to be two large peat plateaus. The northernmost of the four palsas (hereafter 'Peera palsa') has been monitored in this research (Fig. 1c). The palsa consists of a main body with a diameter of around 50 m, the highest point of which was 2 m above the surrounding mire in 2021, and two lower parts, which increase palsa's core-to-edge ratio. The area of palsa was around 1500 m$^2$ in 2021. The vegetation cover on the palsa (Fig. 1d) varies

from almost bare peat surface to dwarf birch shrubs of up to 50 cm in height. The average peat depth around the palsa is 1.3 ± 0.3 m (± SD).

## 2.2 Laassaniemi

The Laassaniemi palsa mire is located between Ala-Kilpisjärvi Lake and the main road (68°56'39" N, 20°54'40" E, 475 m a.s.l.) and is ca. 0.7 ha. The mire includes four palsas, which are disintegrated parts of a larger peat plateau, and all of which are < 1 m high (Fig. 1e). The detailed investigation focuses on the southern palsa (hereafter 'Laassaniemi palsa'), which has over ten times smaller area (ca. 120 m$^2$) than the Peera palsa. The Laassaniemi palsa is more oval-shaped and it was around 18 m long and 7 m wide in 2021. The sparse vegetation on the palsa consists mainly of lichens, cloudberries, and crowberries. The thermokarst pond between the palsas is partly revegetated by sphagnum mosses and common cotton grass. Low (< 20 cm) dwarf birches occupy the dry areas next to the palsa (Fig. 1f). The peat depth around the palsa varies between 1.5 m and 2.8 m.

## 3 Data and methods

### 3.1 RTK-GNSS and active layer measurements

The ALT monitoring sites were established at the end of August 2007. Palsa surface and edges were surveyed using the RTK-GNSS (Topcon Hiper Pro, II, and V) with 1 cm horizontal and vertical accuracies. Permanent control points established by the Finnish Geodetic Institute were used for the local RTK-GNSS base station installation. A 1 m long steel rod was used to measure the depth of the active layer. In total, 515 points were measured at Peera (Fig. 2a), of which ca. 180 were placed every 2 m along 11 transects covering half of the palsa, indicated by the yellow rectangle in Fig. 1c. In Laassaniemi, 374 points were measured (Fig. 2c), of which ca. 240 were placed every 1 m along 18 transects. In the following years (2008–2021), the number of measured points and length of transects varied because many of the points were abolished after years of no permafrost (ALT > 1 m) observations and/or due to the collapse of the palsas. The measurements were done during the last week of August, except in 2013, when the sites were visited on the 9[th] and 10[th] of September (Table S1). In 2018, the active layer was not measured at exactly the same points as in other years due to an RTK-GNSS malfunction. Therefore, we interpolated the 2018 ALT values using the Natural Neighbour method and extracted the values from the interpolated raster layers at the locations of points in 2019.

The Natural Neighbour interpolation of the elevation information was used to create Digital Terrain Models (DTMs) from each year's RTK-GNSS data. We determined the edges of the Laassaniemi palsa and the western half of the Peera palsa for each year based on these DTMs and the ALT values. We then subtracted the base altitude (459.7 and 474.3 m a.s.l. for Peera and Laassaniemi, respectively), representing the water level next to the palsa, from the DTMs to analyse the changes in palsa

height and volume. ArcGIS Pro (ESRI, Inc.) and *terra* package (v1.7-3; Hijmans, 2023) in R (R Core Team, 2021) were used for geospatial data processing and all statistical analyses were done in R.

The ALT trends were analysed by fitting linear regression models using the least-squares method. Only ALT values $\leq 1$ m
were included in the statistical analyses. To capture seasonal thaw dynamics with minimal effects of lateral thermal fluxes and thermokarst at the edges of the palsas, we delineated the areas not visibly affected by lateral permafrost degradation. These top-of-palsa (TOP) areas were 309 m$^2$ at Peera and 49 m$^2$ at Laassaniemi (Fig. 2). The focus of our analyses and results is on the ALT values within these TOP-areas (ALT$_{TOP}$), while the results using all ALT values $\leq 1$ m are presented in the Appendices (Fig. A1 and Table A2). The number of ALT values $\leq 1$ m (all and within TOP) for each year are presented in Table S1.

A correction for the thaw subsidence has been suggested for the ALT observations in ice-rich permafrost areas where the subsidence is slow and spatially uniform (Shiklomanov et al., 2013; Streletskiy et al., 2017). Despite the permafrost thaw in palsa mires being relatively fast compared to the areas described by Shiklomanov et al. (2013) and Streletskiy et al. (2017), we also tested whether using subsidence-corrected ALT$_{TOP}$ values would improve the model fits. The correction for subsidence was done by calculating the sum of the annual mean ALT$_{TOP}$ and total surface subsidence within the TOP compared to the
2007 level.

### 3.2 Aerial and UAS time series

We used a combination of aerial photography and UAS orthomosaic time series to assess the changes in the palsa extent between 1959 and 2021. Photographs from 1959 and 1985, and orthophotos from 2000 and 2012 with a spatial resolution of 50 cm were accessed through the National Land Survey of Finland (NLS, 2022) (Table S2). The UAS data have been collected
with various unoccupied aerial vehicles every August since 2016 and every second June since 2018. In this work, we used data collected in August (Table S3), except for the delineation of palsa areas in 2018, for which June data were used due to better accuracy. Ground control points (GCPs) measured with the RTK-GNSS connected to the local geodetic control points or using Virtual Reference System (VRS) were available for at least one UAS survey per site per year except for Laassaniemi in 2020. The data were processed into orthomosaics and DEMs in Agisoft Metashape software (version 1.8.4), by using the structure
from motion approach (Westoby et al., 2012). Aggressive depth filtering was applied when processing the UAS data into a dense point cloud to minimise the effects of small shrubs or shrub patches. In some parts of the study areas the vegetation cover was too dense to distinguish the ground surface from the point cloud. Therefore, the DEMs created are Digital Surface Models (DSMs). The horizontal resolution of the orthomosaics and DSMs is ~2 cm and ~5 cm, respectively.

To quantify the lateral degradation of permafrost, palsa edges were delineated by visual interpretation of the aerial photographs
and orthomosaics. Distinguishing palsas from old, panchromatic photographs is challenging, especially in shadowed areas. Therefore, the extent of palsas at the beginning of the time series is more ambiguous than in more recent years. In uncertain cases during delineation, the edge resulting in the lesser palsa area was prioritized to avoid overestimation. The time series

was divided into four periods: I) 1959–1984, II) 1985–1999, III) 2000–2011, and IV) 2012–2021. The average annual area loss rate (% a$^{-1}$) with consideration of the annual change in the area was calculated for each period using the following equation adapted from Olvmo et al. (2020):

$$\text{Annual area loss rate} = \left[ \left( \frac{A_{start}}{A_{end}} \right)^{\frac{1}{Y_{start} - Y_{end}}} - 1 \right] \times 100 \qquad (1)$$

where $A_{start}$ and $A_{end}$ are the total area of palsas at the start year ($Y_{start}$) of the respective period and at the end of it ($Y_{end}$).

UAS DSMs, resampled to a 10 cm cell size, were used to analyse height and volume changes (2016–2021) of the two main palsas, from which also the ALT and RTK-GNSS data were collected. Change analysis based on UAS DSMs is sensitive to small changes in vegetation structure and differences in lightning conditions. No notable changes in the vegetation cover of the palsas were observed during the annual visits to the sites by the authors. However, it is possible that small differences in vegetation cover because of, for example, trampling or vegetation growth may have affected the height and volume changes derived from the DSMs. Variations in the UA systems (Table S3), settings used in the data collection and the devices used to collect the coordinates of the GCPs resulted in discrepancies in the DSMs of different years. Due to the lack of stable surfaces unaffected by the "edge-effect" in the UAS data, we were unable to correct for potential global offsets between the DSMs. Therefore, we used the palsa polygons as delineated from the 2016 orthomosaic to extract only the areas of the main palsas from the 2016 and 2021 DSMs. We then used the minimum value within that area as the base altitude for the respective year.

We used two methods to assess the results of height changes based on the 2016 and 2021 UAS DSMs. First, we used TOP polygons to extract the mean 2016 and 2021 heights from the RTK GNSS DTMs and UAS DSMs and compared these mean height values. Second, we used all RTK-GNSS measurement points to extract the absolute elevation (m a.s.l.) and height values to these points from the UAS DSMs and RTK-GNSS DTM heights. We then calculated root mean square errors (RMSEs) and median differences between the elevation values and height values of the two datasets.

### 3.3 Climate data and statistical analysis

The meteorological data in Kilpisjärvi were obtained from the official weather station of the Finnish Meteorological Institute (FMI, Fig. 1b). The data set includes daily values of precipitation (mm), snow depth (cm) and mean air temperature (°C) from 1960 to 2021 as well as wind speed (m s$^{-1}$) and wind direction (°) from 1998 to 2021.

We calculated several climatic parameters known to affect the active layer and/or permafrost degradation (Meier, 2015). These parameters are (square root of) thawing and freezing degree days (TDD and FDD), mean annual, summer and winter air temperatures (MAAT, MSAT and MWAT), and precipitation: total precipitation, rain (precipitation when air temperature $> 0$ °C), snow (precipitation when air temperature $< 0$ °C), and rain during preceding autumn, spring and summer. We also

examined the effects of wind speed during winter, the number of days with particularly low (< -10 °C) and warm (> 15 °C) air temperatures and the number of days with over 10 cm snow depth. We used the hydrological year starting from the beginning of September and ending at the end of August for all annual average values. TDD was calculated as the sum of positive daily temperatures from the start of the calendar year until the day of measurement of the active layer (for the comparison with the ALT values) or until the end of August (for the comparison with annual area loss rates). Full lists of the climatic parameters used in the analyses are presented in Tables 6 and A1.

We also extracted the timing (day of year) and duration of seasonal snow cover, maximum and end of March snow depth and the fraction of precipitation as snow. Snow accumulation is likely lower, and snow melts earlier on wind-exposed palsas than at the weather station. As we were not able to validate snow onset and melt dates from freely available optical satellite data, due to frequent cloud cover, we assumed that the differences in the timing of snow cover between the station and the palsas are similar every year. However, to account for thinner snow cover on the palsas, we estimated the local snow depths at our sites using SnowModel (Liston and Elder, 2006). Meteorological observations from the Kilpisjärvi station, UAS DSMs from 2018 resampled to 30 cm and vegetation classification maps were used as input parameters for the model. The average top-of-palsa values were extracted from the daily snow depth raster layers produced by SnowModel and used to access local snow depth at the end of March (2008–2021). A comparison of the snow depth values by SnowModel and field observations from 2019 (Table S4) showed RMSEs of 8–25 cm, while the RMSEs between field observations and the values at the Kilpisjärvi station were 16–48 cm. A more detailed description of SnowModel and its application to estimate the snow depth on Peera and Laassaniemi palsas is presented in the Supplement.

We used linear regression to assess temporal trends in the climatic parameters and to find which of the parameters influence the ALT, RTK-GNSS–based volume loss and annual area loss rate. For the analysis of climatic drivers of the lateral palsa degradation, we used average values for the same four time periods as in Sect. 3.2. The regressions of temporal trends and correlations were considered statistically significant if $p \leq 0.05$.

## 4 Results

### 4.1 Active layer thickness

The ALT fluctuates annually (Figs. 3 and A1, Table S1). At both palsas, the number of values over one metre has increased, especially at the edges (Figs. 2b and 2d). During the study period of 2007–2021, the average (± SD) $ALT_{TOP}$ at Peera was 60 ± 4.6 cm. A linear trend suggests a moderate deepening of the thawed layer of ca. 0.3 cm $a^{-1}$. The trend, however, is not statistically significant ($R^2 = 0.07$, $p = 0.353$), and the $ALT_{TOP}$ has decreased after reaching ca. 70 cm in 2014. At Laassaniemi, the average $ALT_{TOP}$ was 49 ± 6.5 cm, and the linear trend indicates a significant decrease in the thawed layer at the rate of ca.

1.2 cm a$^{-1}$ (R$^2$ = -0.68, $p$ < 0.001). Linear trends 2007–2021 using all ALT values ≤ 1m show similar results (+ 0.4 cm a$^{-1}$ at Peera, -1 cm a$^{-1}$ at Laassaniemi), although with higher standard deviations (Fig. A1 and Table S1).

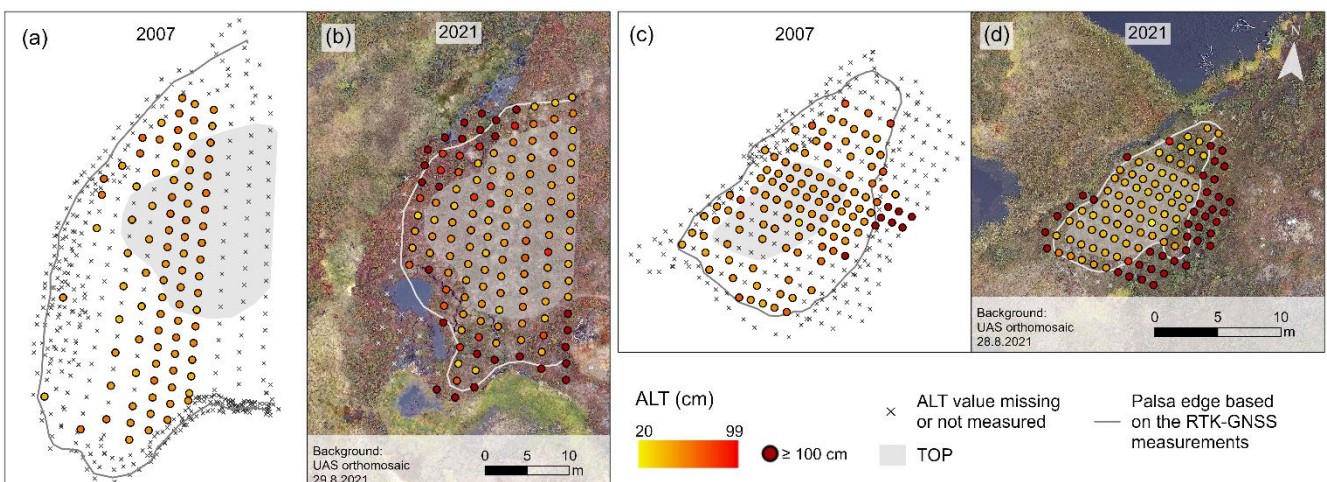

**Figure 2.** RTK-GNSS measurement points and the active-layer thickness (ALT) in 2007 and 2021 at Peera (**a** and **b**) and Laassaniemi (**c**
and **d**). The shaded area indicates the top-of-palsa (TOP) area used for analysing changes in the active layer during the study period.

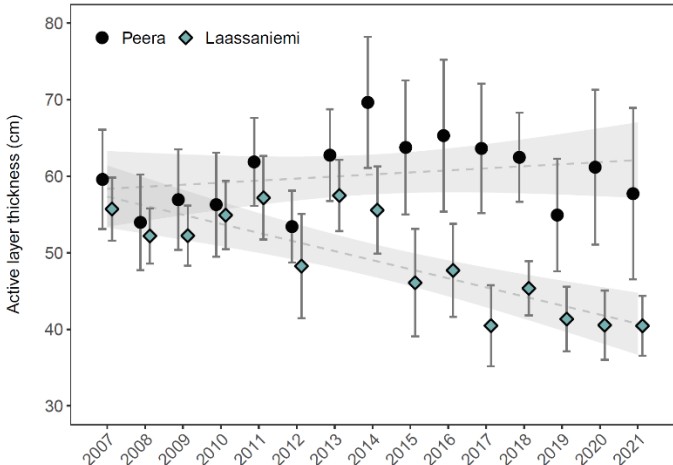

**Figure 3.** Mean (± 1 SD) top-of-palsa ALT 2007–2021. Dashed lines indicate linear trends, and shaded areas represent 95 % confidence intervals.

## 4.2 Climatic drivers of the active layer thickness

Most of the correlations between climatic parameters and the $ALT_{TOP}$ were not significant. Therefore, we focused here only on those correlations, which are comparable with other studies, and/or which showed relatively strong correlations at least at one of the sites. The full list of $R^2$ and $p$-values of the regression analyses is presented in Table A1.

The linear trends (Figs. 4a and 4b) showed no statistically significant relationship between $ALT_{TOP}$ and mean summer air temperature (Peera $R^2 = 0.23$, Laassaniemi $R^2 = 0.03$) or $\sqrt{TDD}$ (Peera $R^2 = 0.16$, Laassaniemi $R^2 = 0.19$). Number of days with air temperature $> 15\ °C$ had significant correlation with the $ALT_{TOP}$, but only at Peera (Fig. 4c, $R^2 = 0.31$). The analysis showed no relationship between winter air temperatures and $ALT_{TOP}$ at Peera (MWAT $R^2 = -0.03$, $\sqrt{FDD}$ $R^2 = 0.03$), and some correlation, although statistically not significant, at Laassaniemi (MWAT $R^2 = -0.25$, $\sqrt{FDD}$ $R^2 = 0.25$) (Figs. 4d and 4e). The direction of these correlations at Laassaniemi suggests that the active layer is shallower after milder winters.

The amount of precipitation affected the $ALT_{TOP}$ only at Peera, and the correlation was significant only for $ALT_{TOP}$ and the sum of rainfall during September and October of the previous year (Fig. 4f, $R^2 = 0.28$). Timing of the snow cover onset (Fig. 4g) had a significant negative correlation at both sites (Peera $R^2 = -0.29$, Laassaniemi $R^2 = -0.27$), while the thickness of the snow cover (Figs. 4h and 4i) showed correlation with the $ALT_{TOP}$ only at Peera (snow depth at the end of March $R^2 = 0.33$, maximum snow depth $R^2 = 0.32$, n. of days with snow depth $> 10$ cm $R^2 = 0.37$). The correlation at Peera was greater with the end of March snow depth values measured at the weather station than with the local snow depth estimated using SnowModel ($R^2 = 0.10$).

Using all ALT values $\leq 1m$ instead of $ALT_{TOP}$ in the regression analyses slightly improved the correlations with some of the precipitation-related parameters at both sites (Table A2). The improvements in the correlations were not high enough to change the $p$-value from $> 0.05$ to $\leq 0.05$, for any of the parameters at either site, however. The $p$-value increased over the 0.05 threshold for the number of days with air temperature $> 15\ °C$ and for rainfall during September and October of the previous year for Peera, and for snow cover onset at Laassaniemi when all ALT values $\leq 1m$ were used compared to $ALT_{TOP}$.

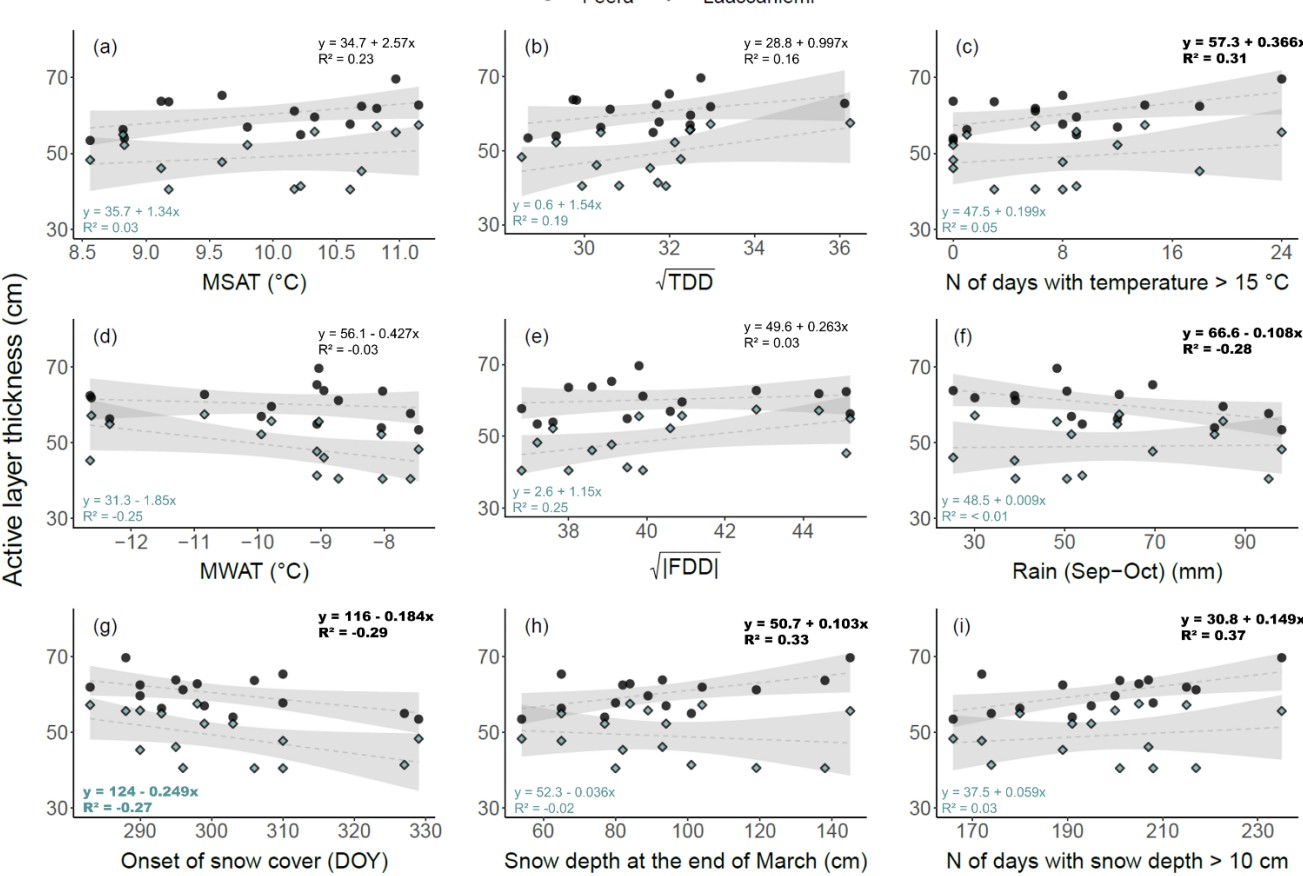

**Figure 4.** ALT$_{TOP}$ vs. different climatic parameters: (**a**) Mean Summer Air Temperature (JJA), (**b**) square root of Thawing Degree Days, (**c**) number of days when the air temperature was above 15 °C, (**d**) Mean Winter Air Temperature (NDJFM), (**e**) square root of the absolute value of the Freezing Degree Days, (**f**) sum of precipitation when air temperature was > 0 °C during September and October in the previous year, (**g**) Day of year (DOY) of the snow cover onset, (**h**) snow depth at the end of March and (**i**) number of days when snow depth was over 10 cm, measured at the Kilpisjärvi weather station. Dashed lines indicate linear trends, and shaded areas represent 95 % confidence intervals. Equations and $R^2$ values of the statistically significant ($p \le 0.05$) correlations are in bold.

The regression analyses of volume loss in relation to climatic parameters did not show any statistically significant correlations (Table A1). The correlations of volume loss with highest $R^2$ at Peera were related to winter air temperatures (MWAT $R^2$ = 0.22, √FDD $R^2$ = -0.22). At Laassaniemi, the volume loss had the highest $R^2$ with the duration of snow cover ($R^2$ = 0.17). The subsidence-corrected ALT$_{TOP}$ had also no statistically significant correlations with the climatic parameters. The correlation

with highest $R^2$ at Peera was with the end of March snow depth ($R^2 = 0.19$) and at Laassaniemi with the mean summer air temperature ($R^2 = 0.13$).

## 4.3 RTK-GNSS-based palsa area, height, and volume changes

Degradation of permafrost in the form of palsa area and height loss is noticeable at both palsas (Fig. 5), but has been relatively stronger at Laassaniemi than at the western half of the Peera palsa (Table 1). At Peera, most of the degradation has been taking place in the southwestern corner. The volume decreased by 55 % within the area covered by the RTK-GNSS surveys between 2007 and 2021. At Laassaniemi, the palsa has degraded from all sides, especially from its southern and northeastern edges. The total subsidence between 2007 and 2021 was double at Laassaniemi (ca. 60 cm) compared with Peera (ca. 30 cm), and the total volume loss was almost 80 %. The annual subsidence increased at Peera and decreased at Laassaniemi over the last ca. 8-10 years of the monitoring period (Table S5 and Fig. S4).

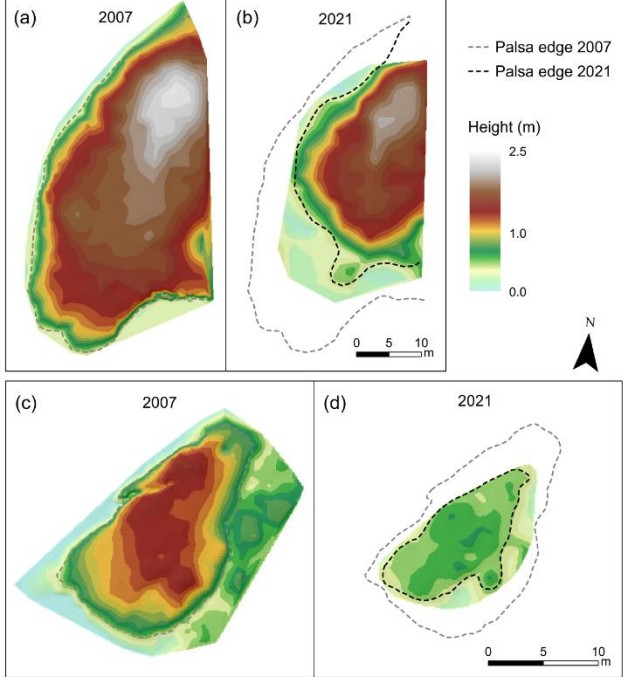

**Figure 5.** Digital terrain models of the western half of the Peera palsa (**a** and **b**) and the Laassaniemi palsa (**c** and **d**) in 2007 and 2021 based on the RTK-GNSS measurements of the surface. Palsa edges are delineated from the RTK-GNSS and active layer measurements. Base altitudes used for the height are 459.7 and 474.3 m a.s.l. for Peera and Laassaniemi, respectively.

**Table 1.** Palsa mean height, area, and volume changes in 2007–2021, from the whole RTK-GNSS measurement area of the years in question. Note that the RTK-GNSS surveys cover only the western half of the Peera palsa.

| | Peera | | | | Laassaniemi | | | |
|---|---|---|---|---|---|---|---|---|
| | 2007 | 2021 | change | % | 2007 | 2021 | change | % |
| Mean height (m) * | 1.6 | 1.4 | -0.3 | -16.5 | 1.2 | 0.6 | -0.6 | -48.9 |
| Area (m²) | 943.8 | 505.9 | -437.9 | -46.4 | 191.9 | 82.5 | -109.4 | -57.0 |
| Volume (m³) * | 1530.6 | 685.2 | -845.4 | -55.2 | 224.4 | 49.4 | -175.0 | -78.0 |

\* Base altitudes used for height and volume calculation: 459.7 and 474.3 m a.s.l. for Peera and Laassaniemi, respectively

## 4.4 Area, height, and volume changes based on UAS DSMs

Based on the UAS DSMs, the mean height of both palsas decreased by 20 cm between 2016 and 2021 corresponding to ca. -17 % change at Peera and ca. -32 % change at Laassaniemi (Table 2). Overall, the lateral degradation was greater at Peera, while the relative volume change at both palsas was similar at ca. -33 % to -38 %. The 2016–2021 DSM difference shows that the northeastern part of the Peera palsa has been relatively stable compared to the southwestern half and the edges, where the thermokarst is the most active (Figs. 6a and 6d). At Laassaniemi, the surface subsidence appears to be lowest at the southwestern side and gradually increases towards the northeastern edge of the palsa (Figs. 6b and 6e). The apparent increase in the height at the northern edge of the Peera palsa is likely a result of differences in how shrubs (Fig. 6c) and their shadows influenced the point cloud generation from the RGB data, and consequently the DSMs. At the northern edge of the Laassaniemi palsa, the increase in height was caused by the rapid expansion of common cotton gras within the adjacent thermokarst pond (Fig. 6f).

Over 30 % larger area of the Laassaniemi palsa as delineated from UAS DSM (122.2 m²) compared to RTK-GNSS and ALT measurements (82.5 m²) is caused by the difference in how the extent of the palsa is defined using these two methods. In the RTK-GNSS and ALT approach, the information about the active layer affected the delineation of the palsa edge, especially in the areas where palsa edge cannot be distinguished based on the topography alone. For the UAS-based delineation, the ALT-values were not used, and the palsa extent is therefore only based on the information about the surface topography and vegetation cover.

**Table 2.** Palsa mean height, area, and volume changes in 2016–2021 based on the UAS DSMs.

| | Peera | | | | Laassaniemi | | | |
|---|---|---|---|---|---|---|---|---|
| | 2016 | 2021 | change | % | 2016 | 2021 | change | % |
| Mean height (m) | 1.3 | 1.1 | -0.2 | -16.7 | 0.7 | 0.5 | -0.2 | -32.1 |
| Area (m²) | 1847.5 | 1483.9 | -363.5 | -19.7 | 133.0 | 122.2 | -10.8 | -8.1 |
| Volume (m³) | 2399.3 | 1605.3 | -794.0 | -33.1 | 91.2 | 56.9 | -34.2 | -37.6 |

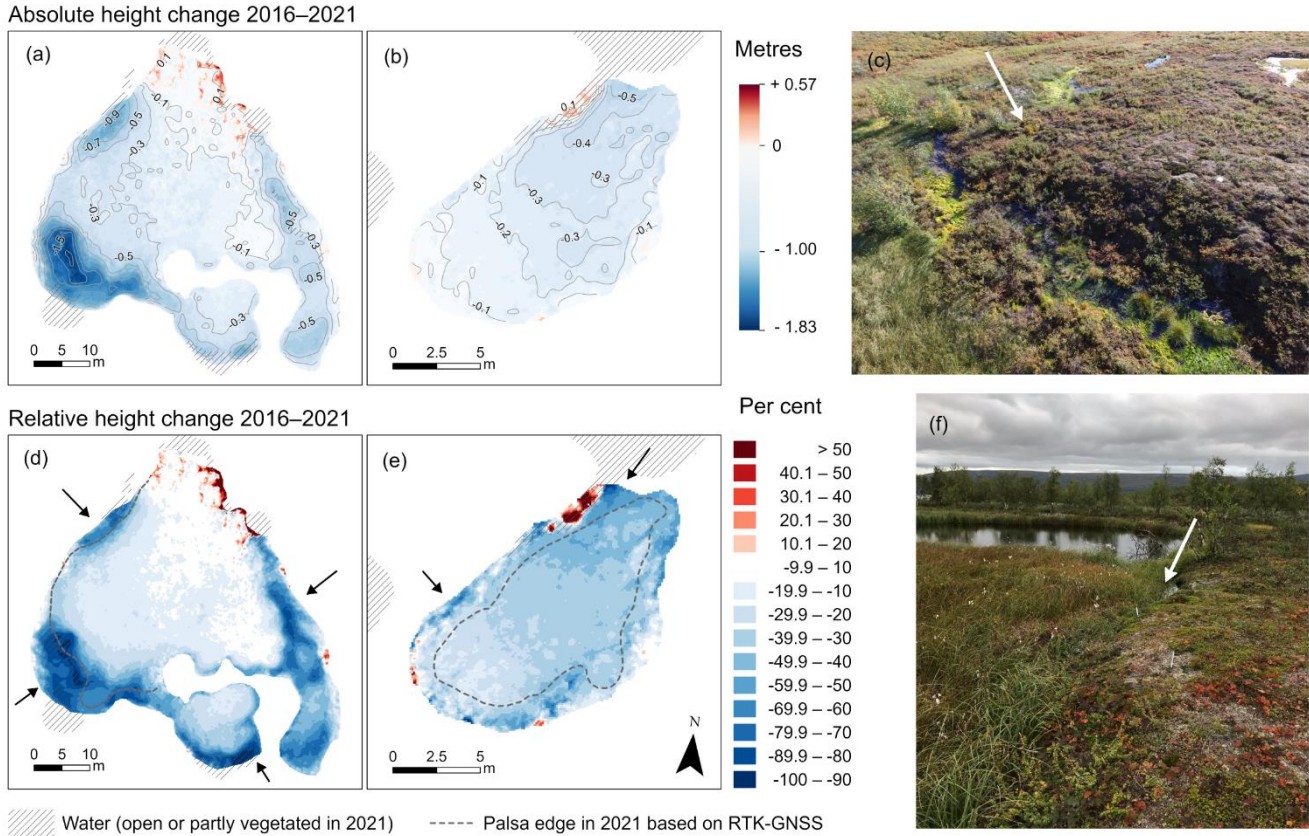

**Figure 6.** Absolute and relative palsa height changes at Peera (**a** and **d**) and Laassaniemi (**b** and **e**) based on UAS DSMs from 2016 and 2021. The effect of vegetation on the DSMs shows as increased values, especially at the northern edge of the Peera palsa indicated by the white arrow in (**c**), whereas the increased values at the northern edge of the Laassaniemi palsa are due to expansion of vegetation within the thermokarst pond indicated by the white arrow in (**f**). Black arrows in (**d**) and (**e**) point to the areas of the most active thermokarst. The dashed line shows the edge of palsas as delineated based on the RTK-GNSS and active layer measurements. The noticeable difference in Laassaniemi palsa area based on the RTK-GNSS compared to the area based on the UAS data is a good example of how the delineated palsa area can vary depending on the chosen method. (Photos by P. Korpelainen, September 2017 (**c**) and T. Kumpula, August 2022 (**f**))

## 4.5 Comparison of RTK-GNSS and UAS 2016–2021

A comparison of the 2016 mean heights within the TOP areas derived from RTK-GNSS DTMs and UAS DSMs showed a 13 cm difference at Peera and only a 2 cm difference at Laassaniemi (Table 3). These differences roughly correspond to the vegetation heights within the TOP areas of the palsas: 0–30 cm at Peera and 0–10 cm at Laassaniemi. The 2021 UAS DSM resulted in lower heights than the RTK-GNSS DTMs at both sites. Therefore, the TOP subsidence is 17 cm (54 %) larger at Peera and 13 cm (48 %) larger at Laassaniemi based on the UAS data compared to the RTK-GNSS.

The RMSEs and median differences using all RTK-GNSS measurement points overlapping with the palsa UAS DSMs (Table 4) also showed larger differences in elevation between the two datasets in 2021 than in 2016. The locations of 2021 GCP-points at both sites were measured with a VRS RTK-GNSS device, while the RTK-GNSS devise used to monitor palsa surface was connected to the local geodetic control points. For the height values, the difference was less pronounced, as the base altitudes for the UAS DSM heights were derived separately for 2016 and 2021 using the minimum elevation value within the palsa area (see Section 3.2).

**Table 3.** Mean heights (m) within top-of-palsa areas derived from the RTK-GNSS DTMs and UAS DSMs from 2016 and 2021.

|  | 2016 | 2021 | change |
|---|---|---|---|
| Peera RTK-GNSS | 1.85 | 1.71 | -0.14 |
| Peera UAS | 1.98 | 1.67 | -0.31 |
| **Difference** | **0.13** | **-0.04** | **-0.17** |
| Laassaniemi RTK-GNSS | 0.78 | 0.64 | -0.14 |
| Laassaniemi UAS | 0.80 | 0.53 | -0.27 |
| **Difference** | **0.02** | **-0.11** | **-0.13** |

**Table 4.** RMSEs (m) and median differences (m) between the RTK-GNSS and UAS DSM elevation (m. a.s.l.), and between the height values derived from the RTK-GNSS DTMs and UAS DSMs from 2016 and 2021.

|  | RMSE elev. | Median elev. diff. | RMSE height | Median height diff. |
|---|---|---|---|---|
| Peera 2016 | 0.09 | -0.02 | 0.16 | 0.14 |
| Laassaniemi 2016 | 0.18 | 0.14 | 0.07 | 0.01 |
| Peera 2021 | 0.26 | -0.24 | 0.11 | -0.03 |
| Laassaniemi 2021 | 0.52 | -0.53 | 0.11 | -0.12 |

## 4.6 Palsa area change 1959–2021

Based on the manual delineation of palsa area from the aerial photography, the overall extent of palsas decreased from 2.18 and 0.39 ha in 1959 to 0.50 and 0.04 ha in 2021 at Peera and Laassaniemi, respectively (Fig. 7). The total palsa area loss between 1959 and 2021 has been -77.3 % at Peera and -89.7 % at Laassaniemi. The average loss rates for the whole study period are -2.4 % $a^{-1}$ at Peera and -3.6 % $a^{-1}$ at Laassaniemi. Contrary to the accelerating degradation at Peera towards the end of the time series, the degradation rate at Laassaniemi has been the lowest during the 2012–2021 period (Fig. 8). No signs of anthropogenic disturbance, such as artificial draining or excavation, are visible in the aerial data.

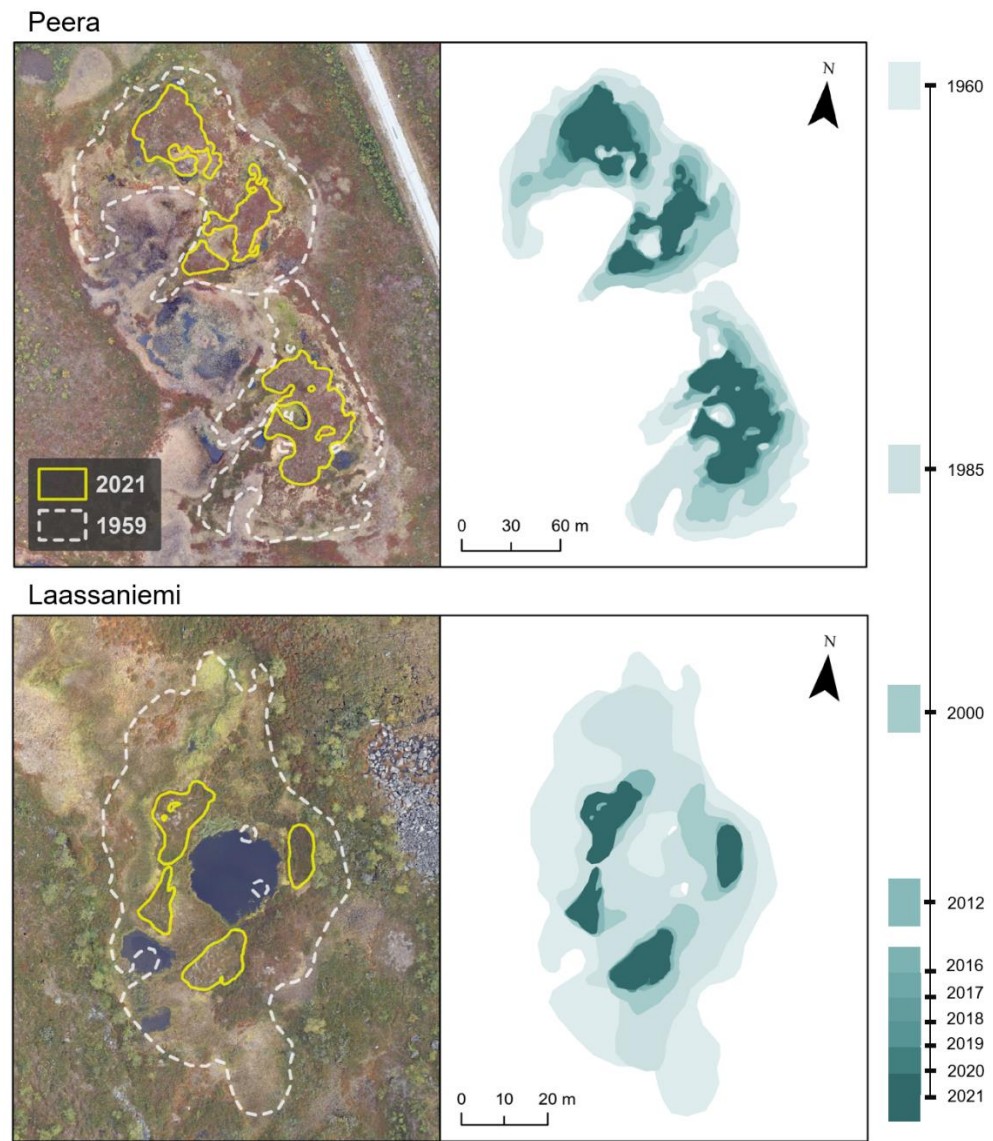

**Figure 7.** Palsa area delineated by visual interpretation of aerial (1959–2012) and UAS (2016–2021) data. **Left:** Palsa edges of 1959 and 2021 overlain on the 2021 UAS orthomosaics. **Right:** Palsa extent in the years having available data. Note that the scales are different for the two sites.

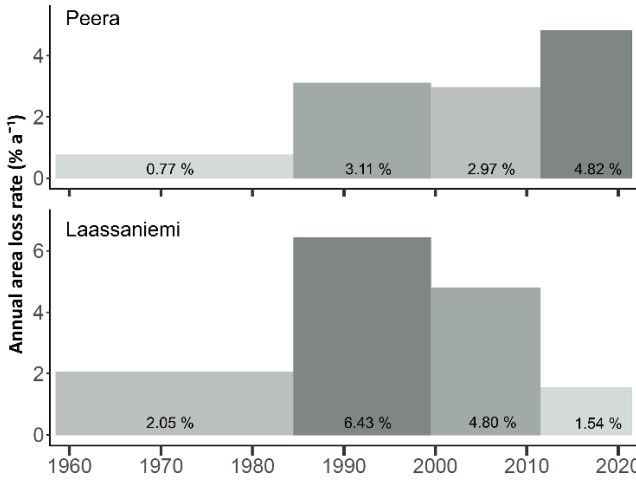

**Figure 8.** Annual area loss rates (% a$^{-1}$) of palsa area within the Peera and Laassaniemi palsa mires based on digitized palsa edges from aerial photographs (1959, 1985, 2000, 2012) and the UAS orthomosaics (2021).

## 4.7 Climate trends and lateral palsa degradation

The climatic averages for each of the four periods are summarized in Table 5. The mean annual air temperature increased significantly over the past 60 years by ca. 0.36 °C per decade. The average air temperatures have warmed especially after 340 1988, and the increase has generally been stronger in the winter months than in the summer. The precipitation has also increased significantly by ca. 25 mm per decade. The period 2000–2012 had the highest amount of precipitation as rain, while the highest average amount of snow was between 1985 and 2000. The snow cover duration was also on average the longest in 1985–2000 and has been decreasing since then.

The full list of R$^2$ and $p$-values of the correlations between the annual area loss rates and climatic parameters is presented in 345 Table 6. Because of the low number of samples (only four periods), these results are only indicative. None of the correlations were statistically significant at 95 % confidence level. At Peera, three parameters related to the winter air temperatures had the highest R$^2$ and $p$-values < 0.1. These parameters were MWAT (R$^2$ = 0.88), √FDD (R$^2$ = -0.88), and the number of days with air temperature < -10 °C (R$^2$ = -0.87). At Laassaniemi, the area loss rates had very little correlation with the climatic parameters; the lowest $p$-values were only around 0.3 for the snow cover onset (R$^2$ = -44) and snow cover duration (R$^2$ = 0.46).

**Table 5.** Climate averages for the periods I) 1961–1984, II) 1985–1999, III) 2000–2011, and IV) 2012–2021, and the trends based on the linear regression for 1961–2021. Values are calculated for hydrological years starting from the beginning of September of the previous year until the end of August. Maximum value of each parameter is in bold. DOY – day of year.

| | 1961–1984 | 1985–1999 | 2000–2011 | 2012–2021 | 1961–2021 trend $(10a)^{-1}$ |
|---|---|---|---|---|---|
| MAAT (°C) | -2.6 | -2.2 | -1.3 | **-0.8** | +0.36 *** |
| MSAT (°C) | 8.9 | 9.4 | **10.1** | 10.0 | +0.25 ** |
| MWAT (°C) | -11.6 | -10.9 | -10.3 | **-9.1** | +0.46 *** |
| Total precipitation (mm) | 416.2 | 464.0 | **515.5** | 508.4 | +24.9 *** |
| Rain (mm) | 240.2 | 226.3 | **289.9** | 280.7 | +9.9 * |
| Snow (mm) | 175.0 | **233.2** | 221.2 | 224.8 | +14.3 * |
| Snow fraction (%) | 42.4 | **49.2** | 42.9 | 44.1 | +0.6 |
| Snow cover onset (DOY) | 293.5 | 290.9 | 296.8 | **304.9** | +1.3 |
| Snow cover end (DOY) | 147.0 | **151.7** | 142.9 | 147.3 | -0.3 |
| Snow cover duration (days) | 217.6 | **226.1** | 211.4 | 207.7 | -1.2 |
| Maximum snow depth (cm) | 103.9 | **120.1** | 96.1 | 106.4 | +0.1 |

$* = p \leq 0.05$, $** = p \leq 0.01$, $*** = p \leq 0.001$

**Table 6.** Coefficient of determination and significance of the correlations between climatic parameters calculated for hydrological years 1961–2021 and annual area loss rates derived from the aerial photography time series (1959–2021).

| | Peera | | Laassaniemi | |
|---|---|---|---|---|
| | R² | p | R² | p |
| [a] MAAT (Sep-Aug) | 0.76 | 0.127 | -0.06 | 0.763 |
| [b] MSAT (JJA) | 0.70 | 0.162 | 0.00 | 0.949 |
| [c] MWAT (NDJFM) | 0.89 | 0.059 | -0.08 | 0.717 |
| [d] √TDD | 0.65 | 0.194 | -0.02 | 0.873 |
| [e] √\|FDD\| | -0.88 | 0.064 | 0.01 | 0.882 |
| N. of days air < -10 °C | -0.87 | 0.069 | 0.04 | 0.791 |
| N. of days air > +15 °C | 0.70 | 0.165 | -0.31 | 0.444 |
| Total precipitation | 0.69 | 0.172 | 0.01 | 0.890 |
| Rain | 0.25 | 0.499 | -0.02 | 0.856 |
| Snow | 0.69 | 0.172 | 0.30 | 0.455 |
| Snow cover onset | 0.50 | 0.292 | -0.46 | 0.322 |
| Snow cover end | 0.00 | 0.929 | 0.09 | 0.692 |
| Snow cover duration | -0.19 | 0.561 | 0.47 | 0.314 |
| Snow depth 31.3. | 0.03 | 0.828 | 0.16 | 0.598 |
| Maximum snow depth | 0.02 | 0.850 | 0.19 | 0.564 |
| N. of days snow depth > 10 cm | -0.03 | 0.826 | 0.03 | 0.842 |

a MAAT - Mean annual air temperature
b MSAT - Mean summer air temperature
c MWAT - Mean winter air temperature
d TDD - Thawing degree days (sum of positive daily temperatures from the beginning of the year until the end of August)
e FDD - Freezing degree days (sum of negative temperatures). Absolute value used for square root.

## 5 Discussion

### 5.1 Permafrost degradation 1960–2021

In Fennoscandia, the climatologically most optimal conditions for palsa existence are MAAT -3 – -5 °C and annual precipitation sum of < 450 mm (Luoto et al., 2004a). The MAAT was already -2.6 °C in Kilpisjärvi between 1960 and 1985 (Table 5). Over the past decades, both air temperatures and precipitation have increased significantly, especially during the wintertime. Following these climatic changes, palsas in the Kilpisjärvi area have experienced a rapid and continuous decrease in area over the past sixty years, and no new palsas have formed.

The degradation rates differ between our two sites and between the four time periods. At Peera, the trend of increasing area loss rates follows the increase in air temperatures and precipitation similar to the permafrost peatland near Hudson Bay in

Canada (Payette et al., 2004) and at the Vissátvuopmi palsa mire in Sweden (Olvmo et al., 2020). Based on the volume loss rates derived from the RTK-GNSS surveys, the permafrost degradation at Peera is particularly driven by the increase in winter air temperatures.

At Laassaniemi, the area loss rate peaked during 1985–2000, which is also the period with the longest snow cover duration and highest maximum snow depths. The RTK-GNSS–based volume loss appears to correlate mainly with the snow cover duration as well. Opposite to Peera, the annual area loss rate was lowest during the last period (2012–2021) at Laassaniemi. Palsas at Laassaniemi are generally much lower than at Peera and are surrounded by dwarf birch thickets and tall common cotton grass (Fig. 1f), which increases the possibility of errors in the delineation of palsa edges by visual analysis of the aerial data. Therefore, it is possible that the UAS-based palsa edge delineation overestimates the palsa extent, which is also indicated by ca. 30 % larger palsa area compared to the RTK-GNSS and ALT –based extent (Fig. 6e).

The UAS-based change analysis also showed ca. 50 % larger thaw subsidence within the top-of-palsa areas between 2016 and 2021 than the RTK-GNSS–based change analysis. The difference is most likely a mixed result of the effects of vegetation cover in the UAS DSMs and the lack of stable reference surfaces around palsas, and the uncertainties related to the variations in the UA systems, the number and quality of the GCPs, and the reference settings of the RTK-GNSS devices. These are all potential issues related to the use of UAS RGB data for elevation change detection in permafrost peatlands. Adding fixed GCP points (Fraser et al., 2022) to our sites could improve the interannual comparability of UAS DSMs in the future.

Variability in local conditions, such as microclimate, vegetation cover, hydrology, and palsa size and shape, may cause differences in degradation rates despite a similar regional climate (Borge et al., 2017). The increasing fragmentation and complexity of the palsas' shape in Peera can have a positive feedback effect on the area loss rate, and vice versa. For example, despite only 10–20 km distance from our sites, the degradation rates were almost three times slower at Vissátvuopmi palsa mire (Olvmo et al., 2020), where palsas were considerably larger covering almost 49 ha in 2015. A direct comparison with other area loss rates reported from North America (Payette et al., 2004; Jones et al., 2016; Mamet et al., 2017) and Norway (Borge et al., 2017) is not possible due to the different ways of calculating the average annual change in per cent. However, when calculated using the same method as used here and in Olvmo et al. (2020), the long-term (~ 1950s to 2010s) degradation rates, for instance, at sites D6 and GF in Mamet et al. (2017) and in Goatheluoppal palsa site in Borge et al. (2017), are similar to ours at ca. 2–4 % $a^{-1}$ (see supp. material in Olvmo et al., 2020).

**5.2 Active layer and climate dynamics**

The analysis of the active layer and climate dynamics over the past 14 years revealed two different cases. At Peera, there was no clear trend in the ALT between 2007 and 2021, although the height and area of the palsa decreased annually. The $ALT_{TOP}$ at Peera varied annually following number of particularly warm days with air temperatures > 15 °C, autumn precipitation, snow cover onset and snow depth. The correlation with the snow cover measured at the weather station is uncertain, however,

as the snow accumulation is generally lower on the Peera palsa than at the weather station (Fig. S3). At Laassaniemi, the annual variations in the $ALT_{TOP}$ could be explained only by the differences in snow cover onset and, to a lesser degree, winter air temperatures. The active layer is becoming shallower, while the palsa surface is continuously subsiding and thawing is occurring at the edges. Excluding the areas affected by lateral thermal fluxes did not obscure any significant relationship between the ALT and climate dynamics. On the other hand, it did result in a few more statistically significant correlations compared to regressions using all ALT values ≤ 1m. For example, at Laassaniemi, none of the regressions showed a significant correlation between the active layer and climatic parameters when all ALT values ≤ 1m were used.

The lack of correlation between the ALT and climatic parameters indicates that other factors, such as vegetation cover, hydrology, soil properties or palsa shape, have strong influence on the seasonal thaw at the Laassaniemi palsa. A shallower active layer in lower palsas compared to higher ones in the same area has also been observed elsewhere in northern Finland (Rönkkö and Seppälä, 2003; Seppälä, 2011). Seppälä (2011) attributed the thicker active layer in higher palsas to a longer snow-free period, which allows more time to receive solar radiation and thaw. Therefore, the lack of any significant increase in the ALT at Peera and a significant decrease in the ALT at Laassaniemi could be (partially) attributed to the lowering of the palsas. However, more palsa and peat plateau sites including ALT and surface elevation monitoring as well as local snow cover observations are needed to support these hypotheses.

A comparison of our results to the published active layer records at the geographically close locations in Sweden shows some similarities, but also contrasting trends. Similar to the observations in Tavvavuoma (Sannel et al., 2016), there was little annual variation between 2007 and 2011 and a relatively shallow active layer in 2012. Relatively deep thaw depths in 2013 and 2014 were also observed in Abisko (CALM, 2022). However, the deepening of the active layer in Abisko continued after 2014, whereas at Peera, the ALT returned to similar values as before 2013. And at Laassaniemi, the active layer was even shallower in 2021 than after a particularly cold summer in 2012. Contrary to the results from Sweden (Åkerman and Johansson, 2008; Sannel et al. 2016), the ALT did not have significant correlation with mean summer air temperature and TDD at our study sites.

The strong negative correlation between the ALT and snow cover onset could be due to the longer time of peat saturation, which increases its thermal conductivity (Kujala et al., 2008) and promotes cooling of the ground in late autumn. The timing of snow accumulation has also been shown to have an impact on the ALT at a peat plateau in northern Sweden (Johansson et al., 2013). Warm air temperature in November can delay the snow cover onset, which could explain the surprising negative correlation between the winter air temperatures and the $ALT_{TOP}$ at Laassaniemi. The changes in snow cover duration have been observed in various Arctic regions, and an even stronger shift in snow-on and snow-off dates is expected with the projected climate change (AMAP, 2017). The effects of snow cover timing on the ground's thermal regime and ALT have been so far investigated mainly in continuous permafrost environments (Ling and Zhang, 2003; O'Neill and Burn, 2017; Jan

and Painter, 2020). Hence, the response of discontinuous and sporadic permafrost to changing snow cover duration warrants more research.

Correcting the ALT$_{TOP}$ values for subsidence did not improve the linear model fits with thaw degree days or any other climatic parameters at our sites. This was not surprising as the rate of subsidence at our sites has been around two to six times greater than at the Alaska North Slope (Shiklomanov et al., 2013; Streletskiy et al., 2017) and even more compared to the Mackenzie River delta in Canada (O'Neill et al., 2019). However, as noted by Martin et al. (2019), the permafrost thaw occurring from the bottom may also cause subsidence of palsas without an effect on the ALT. This suggests that the active layer may appear stable due to subsidence also in rapidly degrading sporadic permafrost environments.

Our results imply that using the ALT alone for assessing permafrost conditions can be insufficient in permafrost peatlands and, therefore, underline the importance of high-accuracy surface position measurements at the ALT monitoring sites. The possibility that the active layer may remain stable or become shallower in the inner parts of palsas despite overall permafrost degradation should be considered when analysing ALT records from similar environments and in modelling efforts of biogeochemical, hydrological, and permafrost dynamics of palsa mires.

## 6 Conclusions

The analysis of the aerial photography time series indicates a rapid degradation of the permafrost in north-west Finland over the past 60 years. At the Peera and Laassaniemi palsa mires, the extent of palsa area degraded over 75 % between 1959 and 2021. At the same time, air temperatures and precipitation have increased significantly. The accelerating lateral degradation at Peera coincides with increasing (winter) air temperatures, while the lateral degradation at Laassaniemi was greatest during the period of high winter precipitation in the 1990s.

Although we found no increasing trend for the ALT between 2007 and 2021, permafrost degradation is evident in the form of lowering palsa surfaces and thermokarst development at their edges. The active layer was deeper especially after winters with thick snow cover and summers with high number of very warm days at Peera, while at Laassaniemi, the relationships between ALT and climatic drivers are weak. Thinner active layer at both sites following a later snow cover onset in the preceding autumn indicates the complexity of feedbacks related to climate warming and consequent shifts in snow cover duration in Arctic regions. Furthermore, the decreasing ALT at Laassaniemi despite the overall decrease in palsa area and height emphasises that the active layer thickness alone is not enough to assess the (in)stability of permafrost.

*Data availability.* RTK-GNSS, ALT, and UAS data used in this study are available upon request from the authors. Meteorological data are available through Finnish Meteorological Institute. Aerial photographs and orthophotos are available through National Land Survey of Finland.

*Author contributions.* Initial study design TK and AC. Supervision: TK and BB. Data collection: all authors. Data processing: MV, AS, and PK. Data analysis and visualisation of the results: MV. Discussion of results and conclusions: MV, AS, TK, and BB. Writing the manuscript: MV, with contribution from AS, EL, TK, and BB.

*Competing interests.* The authors declare that they have no conflict of interest.

*Acknowledgements.* This work was supported by the Vilho, Yrjö and Kalle Väisälä Foundation of the Finnish Academy of Science and Letters, EU Horizon 2020 Research and Innovation Programme Grant agreement no. 869471, the Academy of Finland decision no. 330319, and the Erasmus+ staff mobility programme. We thank all students and teachers of the field

courses during 2007–2021 for their contribution to the collection of the field data throughout the years. We also thank Heather Reese, an anonymous reviewer and the editor Hanna Lee for their comments and suggestions, which helped to improve our manuscript.

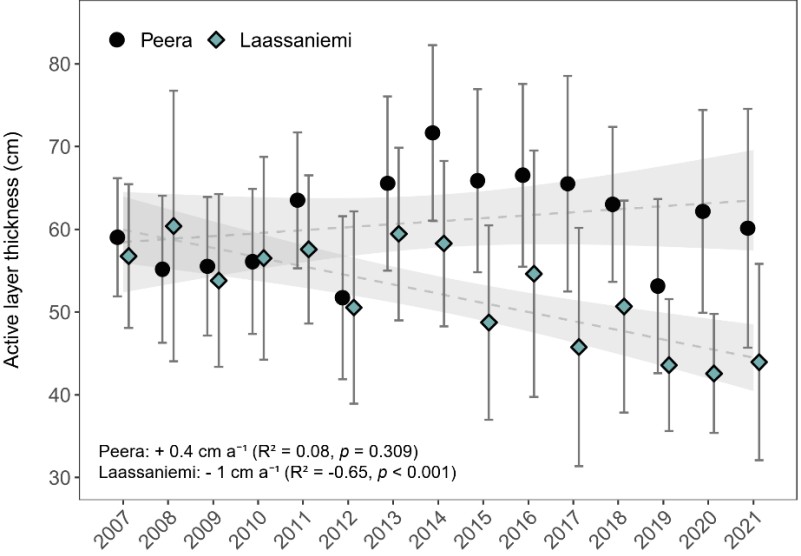

**Figure A1.** Mean (± 1 SD) ALT 2007–2021 using all ≤ 1m ALT values. Dashed lines indicate linear trends, and shaded areas represent 95 % confidence intervals.

**Table A1.** Coefficient of determination and significance of the correlations between climatic parameters and annual active layer thickness ($ALT_{TOP}$), annual subsidence-corrected $ALT_{TOP}$, and annual RTK-GNSS based volume loss in % $a^{-1}$ from the whole RTK-GNSS measurement area of the years in question 2007–2021. Climatic parameters are calculated from the data of the Kilpisjärvi weather station (FMI), except for the "Snow depth 31.3. (SM)", which is based on the values derived using SnowModel. Statistically significant ($p \leq 0.05$) correlations are in bold.

| | Peera | | | | | | Laassaniemi | | | | | |
| --- | --- | --- | --- | --- | --- | --- | --- | --- | --- | --- | --- | --- |
| | $ALT_{TOP}$ | | Sub. Corr. $ALT_{TOP}$ | | Volume Loss | | $ALT_{TOP}$ | | Sub. Corr. $ALT_{TOP}$ | | Volume Loss | |
| | $R^2$ | $p$ | $R^2$ | $p$ | $R^2$ | $p$ | $R^2$ | $p$ | $R^2$ | $p$ | $R^2$ | $p$ |
| [a] MAAT (Sep-Aug) | 0.00 | 0.909 | 0.12 | 0.215 | 0.19 | 0.125 | -0.17 | 0.124 | 0.09 | 0.285 | 0.00 | 0.995 |
| [b] MSAT (JJA) | 0.23 | 0.072 | 0.18 | 0.114 | 0.02 | 0.661 | 0.03 | 0.524 | 0.13 | 0.182 | 0.00 | 0.746 |
| [c] MWAT (NDJFM) | -0.03 | 0.563 | 0.04 | 0.497 | 0.22 | 0.087 | -0.25 | 0.055 | 0.02 | 0.655 | 0.00 | 0.858 |
| [d] $\sqrt{TDD}$ | 0.16 | 0.141 | 0.01 | 0.706 | -0.02 | 0.643 | 0.19 | 0.104 | 0.02 | 0.594 | 0.02 | 0.637 |
| [e] $\sqrt{|FDD|}$ | 0.03 | 0.571 | -0.03 | 0.531 | -0.22 | 0.095 | 0.25 | 0.058 | -0.01 | 0.685 | 0.00 | 0.928 |
| N. of days air < -10 °C | -0.01 | 0.766 | -0.06 | 0.364 | -0.13 | 0.203 | 0.17 | 0.125 | -0.02 | 0.592 | -0.02 | 0.628 |
| N. of days air > +15 °C | **0.31** | **0.033** | 0.14 | 0.172 | -0.01 | 0.702 | 0.05 | 0.439 | 0.09 | 0.266 | -0.02 | 0.634 |
| Total precipitation | 0.01 | 0.774 | 0.02 | 0.575 | 0.05 | 0.462 | -0.10 | 0.245 | 0.00 | 0.736 | -0.02 | 0.620 |
| Rain | -0.07 | 0.325 | -0.02 | 0.584 | 0.00 | 0.920 | 0.00 | 0.899 | 0.00 | 0.894 | -0.05 | 0.453 |
| Snow | 0.14 | 0.166 | 0.10 | 0.260 | 0.05 | 0.436 | -0.08 | 0.300 | 0.02 | 0.656 | 0.00 | 0.763 |
| Rain (May-Jun) | -0.10 | 0.243 | -0.09 | 0.265 | -0.15 | 0.177 | 0.05 | 0.435 | -0.02 | 0.645 | -0.01 | 0.688 |
| Rain (Jul-Aug) | 0.06 | 0.368 | 0.03 | 0.522 | 0.00 | 0.906 | -0.02 | 0.597 | 0.04 | 0.455 | -0.05 | 0.459 |
| Rain (Sep-Oct of previous year) | **-0.28** | **0.042** | -0.10 | 0.256 | 0.10 | 0.273 | 0.00 | 0.912 | -0.10 | 0.248 | 0.00 | 0.936 |
| Snow cover onset | **-0.29** | **0.040** | 0.00 | 0.827 | 0.06 | 0.397 | **-0.27** | **0.047** | 0.04 | 0.463 | -0.14 | 0.180 |
| Snow cover end | 0.02 | 0.610 | 0.00 | 0.730 | 0.06 | 0.409 | -0.02 | 0.585 | 0.00 | 0.818 | 0.02 | 0.613 |
| Snow cover duration | 0.19 | 0.106 | 0.00 | 0.937 | 0.00 | 0.937 | 0.11 | 0.226 | -0.02 | 0.610 | 0.17 | 0.145 |
| Snow depth 31.3. (SM) | 0.10 | 0.271 | 0.11 | 0.241 | 0.03 | 0.540 | -0.04 | 0.506 | 0.05 | 0.461 | 0.00 | 0.929 |
| Snow depth 31.3. | **0.33** | **0.025** | 0.19 | 0.101 | 0.00 | 0.933 | -0.02 | 0.612 | 0.09 | 0.269 | 0.00 | 0.978 |
| Maximum snow depth | **0.32** | **0.027** | 0.14 | 0.162 | 0.00 | 0.974 | -0.01 | 0.693 | 0.05 | 0.411 | 0.03 | 0.547 |
| N. of days snow depth > 10 cm | **0.37** | **0.016** | 0.10 | 0.253 | 0.01 | 0.733 | 0.03 | 0.540 | 0.02 | 0.590 | 0.10 | 0.277 |
| Wind speed (Nov-Apr) | 0.03 | 0.540 | 0.07 | 0.353 | 0.13 | 0.197 | -0.06 | 0.391 | 0.00 | 0.840 | 0.00 | 0.996 |

[a] MAAT - Mean annual air temperature
[b] MSAT - Mean summer air temperature
[c] MWAT - Mean winter air temperature
[d] TDD - Thawing degree days (sum of positive daily temperatures from the beginning of the year until the ALT measurements)
[e] FDD - Freezing degree days (sum of negative temperatures). Absolute value used for square root.

**Table A2.** Coefficient of determination and significance of the correlations between climatic parameters and annual active layer thickness 2007–2021 using all ALT values ≤ 1m. Climatic parameters are calculated from the data of the Kilpisjärvi weather station (FMI), except for the "Snow depth 31.3. (SM)", which is based on the values derived using SnowModel. Statistically significant ($p ≤ 0.05$) correlations are in bold, and the values that are better compared to the results using only the $ALT_{TOP}$-values are in italics.

| | Peera | | Laassaniemi | |
|---|---|---|---|---|
| | R² | *p*-value | R² | *p*-value |
| [a] MAAT (Sep-Aug) | 0.00 | *0.850* | -0.10 | 0.243 |
| [b] MSAT (JJA) | 0.23 | *0.071* | 0.00 | 0.801 |
| [c] MWAT (NDJFM) | -0.02 | 0.643 | -0.17 | 0.129 |
| [d] √TDD | 0.16 | *0.139* | 0.12 | 0.216 |
| [e] √\|FDD\| | 0.01 | 0.667 | 0.15 | 0.148 |
| N. of days air < -10 °C | 0.00 | *0.727* | 0.09 | 0.278 |
| N. of days air > +15 °C | 0.24 | 0.067 | 0.04 | 0.499 |
| Total precipitation | 0.00 | 0.83 | -0.06 | 0.363 |
| Rain | *-0.09* | *0.275* | 0.00 | *0.876* |
| Snow | 0.14 | *0.163* | -0.08 | *0.298* |
| Rain (May-Jun) | *-0.15* | *0.150* | 0.05 | *0.414* |
| Rain (Jul-Aug) | 0.06 | 0.396 | -0.01 | 0.673 |
| Rain (Sep-Oct of previous year) | -0.24 | 0.063 | *0.01* | 0.692 |
| Snow cover onset | ***-0.30*** | ***0.033*** | -0.22 | 0.079 |
| Snow cover end | *0.04* | *0.501* | -0.04 | *0.477* |
| Snow cover duration | *0.24* | *0.066* | 0.06 | 0.399 |
| Snow depth 31.3. (SM) | 0.10 | 0.259 | -0.04 | 0.489 |
| Snow depth 31.3. | **0.30** | **0.034** | *-0.04* | *0.472* |
| Maximum snow depth | **0.30** | **0.034** | *-0.03* | *0.528* |
| N. of days snow depth > 10 cm | ***0.44*** | ***0.007*** | 0.00 | 0.774 |
| Wind speed (Nov-Apr) | 0.02 | 0.594 | *-0.07* | *0.329* |

a MAAT - Mean annual air temperature
b MSAT - Mean summer air temperature
c MWAT - Mean winter air temperature
d TDD - Thawing degree days (sum of positive daily temperatures from the beginning of the year until the ALT measurements)
e FDD - Freezing degree days (sum of negative temperatures). Absolute value used for square root.

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
