# Peer review of "Permafrost degradation at two monitored palsa mires in north-west Finland"

_EGUsphere, 2022_

## Author Comment (AC1)

**Our point-by-point replies (in black) to the question and comments by Dr. Reese (in blue).**

The paper by Verdonen et al., is a study on degradation of palsas and permafrost plateaus over two sites in northwestern Finland, where Active Layer Thickness and ground elevation has been measured systematically and annually since 2007, and aerial photos exist from 1960 and onwards (including UAS images) to measure lateral degradation. Verdonen et al user linear regression to relate changes in palsa area and height changes to climatic parameters. This paper contributes to a better understanding of the changes in the sensitive palsa mires of Fennoscandia, and their association with climate variables. It's well written, and represents a further step in understanding palsa dynamics and associated influences, both climatic and otherwise.

We thank Heather Reese (Reviewer 1) for insightful comments and many good questions, and are grateful for her time and effort in providing valuable feedback. We believe that addressing the issues raised by Dr. Reese will improve our manuscript substantially and will add further insights not only to the permafrost and climate dynamics but also the methodologies used in the monitoring of palsas.

Main issues are marked with a *.

**Questions and comments**

1. Line 4 – It would be informative with a short form of the Latitude after the names.

We will add "68° N" after the site names as suggested.

2. Line 5 – I don't think that is true that your study focuses on the time period covered by your UAS data. It focuses just as much on your ALT measurements from 2007-2021 (albeit only within the non-degraded area of the palsas), as well as your RTK-GNSS ground elevation measurements, which to me is the more interesting data set since it should be more accurate. Even the next line in the abstract mentions using the longer time series of aerial photos. So I would take away this sentence ("The emphasis is on detailed change detection …").

Our initial idea was to focus more on detailed change detection during the last 6 years of the investigated period. However, the results of the ALT and RTK-GNSS measurements were more interesting and shifted the focus of this article. Therefore, we agree with this comment and will rephrase the beginning of the abstract so that there is no emphasis on the UAS data.

3. Line 8 – Mention that the ALT data are annual from 2007-2021.

Done.

4. L61 – name the years that your study is looking at, rather than using the more vague "the investigation period" phrasing.

Thanks to this comment we noticed that the investigation periods are presented only in the next paragraph. Therefore, we decided to rephrase the first research question to include the periods as

follows: "(1) How did the lateral extent of palsas (1959–2021), palsas' height (2007–2021) and the active layer thickness (2007–2021) change during the investigation periods?"

5. L66 – It would be better if you indicated what kind of sensor you are using for data collection from the UAS platform. Is it an RGB camera, an NIR camera, and/or a Lidar sensor?

Thank you for the suggestion. Indeed, many different sensors have been used to collect data with drones at these study sites. However, here we used only the data collected using RGB-sensors. We will edit the text to mention it as follows:

"For the last six years (2016–2021), the UAS **RGB-**data have enabled capturing changes in palsas at ultra-high **(< 5 cm) spatial** resolution."

6. L91 – Describe the area of the two palsas in the same way and give their dimensions, as the shape of the palsa may affect how it reacts.

We will revise the descriptions of the two palsas so that their dimensions and particularly differences in shape and size are more clear. In the revised version, the dimensions of the Peera palsa will be described as follows:

"The palsa consists of a main body with a diameter of around 50 m, and the highest point of which was 2 m above the surrounding mire in 2021, and two lower "extensions", which increase the palsa's core-to-edge ratio. The area of palsa was around 1500 $m^2$ in 2021."

And the dimensions of the Laassaniemi palsa will be described as follows:

"The detailed investigation focuses on the southern palsa (hereafter 'Laassaniemi palsa'), which has over ten times smaller area (ca. 120 $m^2$) than the Peera palsa. The Laassaniemi palsa is more oval-shaped and it was around 18 m long and 7 m wide in 2021."

7. Figure 1 d and f – If these are 1 or 2 m high, are these palsas or peat plateaus? Or were these taller some decades ago? Just double checking, seeing as you made a point about the difference between the two.

We do not have information about the height of Peera palsa before the beginning of the monitoring in 2007, when its highest point was around 2.5 m above the surrounding mire surface. In Laassaniemi, the height of at least one of the palsa mounds was closer to 2 m based on a photograph taken in 1997, indicating that the whole palsa or peat plateau used to be higher than the ca. 1.5 m observed in 2007. Overall, it seems that all palsas in our study areas are disintegrated parts of larger peat plateaus.

8. *In general – I think it would be better if you used the terms Digital Terrain Models (DTMs) and Digital Surface Models (DSMs) instead of the umbrella term DEM, particularly since your article refers to both kinds of elevation models. Or at least conscious use of the terms. Your RTK-GNSS created DTMs while your drone images will create DSMs.

Thank you for pointing this out. Our reasoning to use only the umbrella term "DEM" was to avoid confusion by using too many similar terms. We acknowledge that the result was the opposite, and it is better to state clearly whether we refer to DTMs or DSMs. Therefore, we will change the RTK GNSS-based DEMs to DTMs throughout the text. We will also add information about UAS-based

DEM processing in the first paragraph of Section 3.2 and change UAS DEMs to UAS DSMs throughout the text.

9. *L120 – Do you mean that if there was lateral degradation in any year from 2007-2021 that you did not include this in the ALT measurements used in the regressions? If so, that should have some effect on your result (and maybe this is why you don't see strong relationships between ALT and climate parameters at the larger of the two palsas). Can you motivate your choice and make clear how using only the "Top of Palsa" mean ALT measurement can affect your results in the Discussion section. It seems like you would be missing the bigger changes. You can see the points you are missing when looking at Fig 2.

This is a very good point, and we agree that the effect of using only top-of-palsa (TOP) ALT values could have been motivated and discussed more in the manuscript. Only the values within the delineated TOP-areas were included in the calculation of the mean ALT and regression analyses. Our hypothesis was that we could better capture the relationship(s) between climatic parameters and annual variations in the ALT by excluding the areas most affected by lateral thermal fluxes. Preliminary analyses of all ALT values ≤ 1 m also showed very similar trends to the trends using only the TOP-values, which further justified focusing on only the most "stable" parts of the palsas.

To avoid speculations in the discussion regarding the effects of using only the TOP values, we ran the same regression analyses with the mean ALT values calculated from all ≤ 1 m measurements. The results showed similar temporal trends as TOP-ALT, but with a higher standard deviation. A comparison with climatic parameters showed less correlation with temperature-related parameters and better correlation with precipitation-related parameters when all ≤ 1 m values were used. As we want to refer to these results in the discussion, we decided to include the analyses of all ≤ 1 m values in the manuscript, either as part of the main text or as an appendix.

10. Section 3.2 – More details are needed on the sensors and specifications used to create these data. A Table could be useful here. What camera? What scale (or GSD- ground sampling distance) are the original images taken at? What full date? Which photo dates were the panchromatic, and what were the others? With the UAS, what platform (since this helps indicate which GPS was used)?

We compiled a table indicating the details of the aerial data used as suggested. This table will be added to the Supplementary materials.

While looking for more details about the aerial photographs we found a mistake in the acquisition year of the first photographs. The correct year is 1959, as opposed to 1960, which was indicated in the web platform where we accessed the data. We have corrected this throughout the manuscript and in the calculations of the annual area loss rates and the results of the regression analyses shown in Table A2.

11. *L145 – You listed a number of issues that you ran into. In addition to this, the UASbased data result in elevations that include vegetation heights (DSMs) and are therefore not completely reliable for showing accurate elevation from year to year, and therefore subsidence and volume changes over time. How tall is the vegetation on the palsas? In any case, this should be a primary reason why you can't calculate reliable volume changes from these data. I would reword this section so that this is acknowledged. However, the orthophotos are useful. It will be much better when you get the UAS-Lidar data for calculating volume changes!

Vegetation height varies from zero centimetres (bare peat surface) to ca. 50–60 cm. We agree that the effect of vegetation cover on the UAS-based heights and volumes should be addressed more in the manuscript. Therefore, in the revised version, we will add the uncertainties due to the vegetation cover along with the issues regarding georeferencing (see our reply to the next comment) to the description of the UAS DSMs processing. And yes, including UAS-LiDAR data will (hopefully) improve our ability to detect changes in palsa topography without vegetation issues.

12. L145 – I think you should also indicate how you geo-referenced your UAS data. You mention problems with the equipment.

Since you have RTK-GNSS data taken annually, couldn't you calculate an RMSE for elevation of the UAS DSMs, indicating their potential error? Then again, that would mean you are comparing DTM and DSM. But still, you might be able to observe systematic errors across the UAS DSM. When you mention in results that you see a trend from southwest to northeast (Line 225), I wonder if it is due to a tilting of the UAS DSM, which can easily happen when good georeferencing isn't possible.

My main point here is not to re-do a lot of work or invest a lot more time in the UAS DSMs, because frankly these will always include uncertainty due to 1) including all surface heights and 2) poor geolocation accuracy if not fixed with RTK-GPS control points. I think you just have to admit and realize the weaknesses of that data set for accurately measuring subsidence.

Reviewer 2 also raised the question about the georeferencing and processing of the UAS DSMs. In addition to the lack of consistency regarding the equipment and UAS data collection, the comparison of the UAS DSMs is further complicated by the lack of stable and fixed control points within the study areas. We will clarify this in the revised manuscript. As recommended, we calculated the RMSEs between the RTK-GNSS elevation values and the UAS DSM values for 2016 and 2021.

We have checked the DSMs of the Laassaniemi palsa for potential tilting, as suggested, and compared them to the RTK-GNSS-based DTMs from the same years. The subsidence in the Laassaniemi DTMs is also highest in the northeastern corner and lowest at the southwestern end. Therefore, we ruled out a tilting effect. However, in 2021, the surface of the Laassaniemi palsa was surprisingly higher based on the RTK-GNSS DTM than derived from UAS DSM. This difference, the RMSEs, and further comparison of the palsa changes based on the RTK-GNSS DTMs and UAS DSMs will be presented in new Section 4.5 of the revised manuscript where we compare the changes in the palsas based on the UAS DSMs and RTK-GNSS DTMs.

13. L165 – Again this is a DSM and not a DTM (or DEM), with vegetation included. Finland has a national Lidar scanning – why didn't you use the DTM from that for the snow model (or even better, both)? Too coarse? Can this account for the rather large differences between modelled and the reference snow depth measurements (10-30cm difference)? Also, where was the vegetation classification from? Your own? In any case, what classes were there?

Yes. The 2 m DEM (or DTM) from the National Land Survey of Finland is too coarse in this case, as the idea was to model the snow distribution with considering small-scale variations in the topography.

Vegetation classification of the Peera palsa mire used by A. Störmer (2020) in his Master's thesis was performed by another student (Tomhave, 2018), who mapped the vegetation using UAS orthomosaics and whose results were validated in the field. This classification was then adapted into the classes used in the SnowModel, in which there are predefined vegetation types with associated

snow-holding capacities (see Table 1 in Liston and Elder, 2006). Following SnowModel vegetation types were used with the classes from the classification by Tomhave (2018) in the brackets:

- Erect Shrub Tundra (Dwarf Birch)
- Low Shrub Tundra (Dwarf Shrub)
- Prostrate Shrub Tundra (Lichen)
- Arctic Gram, Wetland (Sphagnum, Peatland vegetation)
- Bare (Bare rock, Peat)
- Water (Water body)

These same classes were also used to classify vegetation types in the Laassaniemi palsa mire by A. Störmer for the use in this work.

To further address this comment and another reviewer's questions regarding the SnowModel, we will include a more detailed description of the SnowModel, the parameters used, and a summary of the results from the thesis by A. Störmer in the Supplementary materials of the revised manuscript.

14. *L170 – I don't think the explanatory parameters are clearly given. A table could help here, or else you could more clearly state it in the text. For example, did you not test any precipitation variable, besides snow?

The explanatory parameters are introduced in the second paragraph of Section 3.3, where we briefly mention that precipitation was calculated for different seasons. In the revised manuscript, we will add more information about the parameters related to precipitation, including total precipitation, rain, snow, and rain during spring/early summer (May-June), summer (July-August) and autumn (September-October). We will also add a reference to Tables A1 and A2, where full lists of parameters used are given.

15. Fig 2 – I found it hard to see the outline of the palsa. Maybe a little thicker. Also, you should mention what your image is in the background of the 2021 images, and what date it was taken.

We have increased the line thickness of the palsa outlines and added dates of the UAS orthomosaics used as the background as suggested.

16. Fig 3 – Very nice information! This figure raises a lot of questions for me, such as What happened in 2012-2014 to cause this change in ALT?. Also, why the divergence in responses between the two palsa sites after 2014? I interpret the large error bars on Peera to indicate the faster degradation in process, likely due to the small size of the palsa, and the high edge-to-core ratio. Do you think the 2014 ALT measurement is correct for Peera? What causes it to be the biggest thaw measurement in case it is correct?

Thank you for these thought-provoking questions. We checked the data again. Almost all 2014 ALT values were higher at Peera than in 2013, despite the earlier measurement dates in 2014. Therefore, we are confident that there are no errors resulting in higher ALT values compared to other years. The top-of-palsa area is 309 m$^2$ at Peera and 49 m$^2$ at Laassaniemi. Larger area and more variability in the vegetation cover allows larger spatial variability in the ALT at Peera, which could explain longer "whiskers" in Figure 3. Based on the regression analyses, the mix of warm summers in 2013 and 2014 and early snow onset in the autumn of 2013 could be the reason for the deeper thaw in 2014. In addition, based on the regression analyses, the ALT at Laassaniemi is less sensitive to the mean summer air temperature or the number of particularly warm (> +15 °C) days, which could explain the lack of similar deep thaw in 2014. After 2014, the trends in the TOP ALTs of the two sites were rather

similar (- 1.5 cm per year at Peera and -1.8 cm per year at Laassaniemi); thus, the divergence seems to have been caused by this much deeper thaw in 2014 at Peera than at Laassaniemi. In the revised manuscript, we will add information about the size of the TOP-areas and include a table in the Supplementary materials showing the measurement dates, mean ALT, and number of observations (all ≤ 1m values and within TOP).

17. Line 191 – Give the R2 value of the few mentioned correlated variables in the text.

Done.

18. Line 212 – I find this paragraph to be confusing due to the mix of observing what I interpret you to mean lateral degradation as well as subsidence. It would be good to be clear here. The heading is about subsidence or volume change with the RTK-GNSS and the top of the palsa measurements. Otherwise, did you use RTK-GNSS to map the area loss (lateral degradation)? It is unclear, due to the heading, and then the mix of different vaguely worded "degradations".

We will add a following clarification in the first sentence of Section 4.3: "Degradation of permafrost **in the form of palsa area and height loss** is noticeable at both palsas... ". We will also edit this section so that it is clearer that we analysed height, area and volume within the areas covered by the RTK-GNSS surveys, not only top-of-palsa.

19. A thought: Since you have measurements in both places, what is the relationship between the RTK-GNSS measured annual subsidence and annual change in ALT? You wouldn't expect (intuitively) to see a fluctuation in ALT at the same time as you have a constant loss of palsa height. Would be a very good figure to include, since you have the data. (OK, I see in the Discussion you mention this, and try to explain it).

A comparison of the annual changes in the top-of-palsa ALT and annual RTK-GNSS-based subsidence did not reveal any correlation between the two variables. In other words, the active layer can be thinner or thicker compared to previous year, regardless of the degree of subsidence. The relationship between subsidence and ALT, on the other hand, is more apparent, especially at Laassaniemi, where subsidence has slowed since 2012. At Peera, the temporal trend in subsidence is the opposite. Further analysis of the annual subsidence is beyond the scope of our article in its current version, but we could add a figure in the results showing the temporal changes in the mean height and depth of the active layer within the top-of-palsa areas.

20. Table 1 indicates that your volume change measurements using your DTM from RTK-GNSS is based only on the "Top of Palsa" area. Good to make sure that is clearly stated in the methods.

This was not our intention. Therefore, to avoid confusion, we will edit Table 1 caption as follows: "Palsa mean height, area, and volume changes in 2007–2021, **derived from the whole RTK-GNSS measurement area of the years in question.** Note that the RTK-GNSS surveys cover only the western half of the Peera palsa."

21. Line 220 – Include in the sentence that this is height change measured by the UAS DSMs. Also, are you measuring only the "Top of Palsa" area, or what area are you using? To try to figure that out, I read back in methods, where it sounds like you have used the 2016 extent of the palsa, as

delineated from the very detailed orthomosaic, so it will be I guess, a different area than "Top of Palsa". Do I interpret that correctly?

Yes. The areas covered by the UAS DSMs differ from the areas covered by the RTK-GNSS DTMs. Unlike the RTK-GNSS surveys, the UAS DSMs cover the investigated palsas completely, which provides better overview of the changes in the palsas, especially at Peera. We will add "**Based on the UAS DSMs**, ..." at the beginning of the section as suggested.

22. A thought: you would be able to confirm whether subsidence of 20 cm between 2016-2021 found using UAS DSMs corresponds with the subsidence measured by RTKGNSS from the same time period 2016-2021, given that you looked at the same area.

We did consider adding the comparison of palsa changes based on the RTK-GNSS DTMs and UAS DSMs early in the process of preparing the manuscript, but left it out from the final version. Based on this comment and the comments of Reviewer 2, such comparison should be included. Therefore, we will add a new Section 4.5, as mentioned earlier in the reply to the comment #12, showing the differences in the mean top-of-palsa heights derived for 2016 and 2021. In this section, we will also include the RMSE for the elevation and height values between the RTK-GNSS and UAS DSMs.

With this comparison, however, we cannot confirm the 20 cm subsidence found in the UAS DSMs as they cover larger areas than the RTK-GNSS surveys (as mentioned in the previous reply). Within the TOP areas and within the overlapping areas, the comparison shows that the mean height change of the UAS DSMs is double that of the RTK-GNSS DTMs. We will add these results and discuss them in the revised version of the manuscript.

23. Line 229: Well, you can't measure the internal permafrost with the RGB images which only show the surficial extent of the palsa. Also Line 294 you refer to how UAS data can lead to overestimation of permafrost. The aerial photos, or any surficial representation of the palsa is only that – the representation of the geomorphological form of the palsa. To find the permafrost, which is an internal characteristic, so far the ALT measurements are needed. Also in Line 294 – it wouldn't be only UAS, but also any aerial photo, or even Lidar that would "overestimate permafrost".

That is true. Our intention was to highlight the difference in the extent of the Laassaniemi palsa derived by the two methods and explain the reason for this difference. We will rephrase this paragraph so that the emphasis is on the fact that we used the ALT values to delineate the palsa extent from the RTK-GNSS DTMs but did not use them for palsa delineation from the UAS data. The revised paragraph will read as follows:

"Over 30 % larger area of the Laassaniemi palsa as delineated from UAS DSM (122.2 m$^2$) compared to RTK-GNSS and ALT measurements (82.5 m$^2$) is caused by the difference in how the extent of the palsa is defined using these two methods. In the RTK-GNSS and ALT approach, information about the active layer affects the delineation of the palsa edge, especially in areas where the palsa edge cannot be distinguished based on topography alone. For the UAS-based delineation, the ALT-values were not used, and the palsa extent is therefore only based on the information about the surface topography and vegetation cover."

We agree that it should be clearer that delineation of palsas from optical aerial data is not the same as delineating the actual permafrost extent, and we will edit the Line 294 to incorporate this point.

24. Line 239/240 – Include in the sentence that this measurement is derived from manual delineation of palsa area from the aerial photos from 1960, …2021.

We will add "**Based on the manual delineation of palsa area from the aerial photography**,…" at the beginning of the paragraph.

25. Line 240 – that's quite a sad loss of area…

Indeed. At these loss rates we will not have our monitored palsas for much longer.

26. Fig 6 – Legend text is pretty small. Also, I was confused about which legend belonged to which square. Maybe better to make the figure a little bigger, and clearly divide the two sides of absolute and relative change maybe with some lines or column names and Legend heading.

We will rearrange Figure 6 so that the absolute and relative changes are now more clearly separated and enlarge the legend text for better readability.

27. Fig 7 – Nice map, I like this a lot. Is there a way to make it larger in the publication?

Thank you! We will increase the size of Figure 7 in the revised manuscript.

28. Line 259 – Much better description of the results is needed here. What was the R2 of the most correlated climatic variables? Without proper description of the result, it is hard to have any discussion, and hard to compare to other studies (eg Olvmo et al. 2020).

We will add the $R^2$-values of the highest correlations and rephrase the paragraph so that it states more clearly the lack of significant correlations and that the regression results are rather indicative because of the low number of samples. We will also move Table A2 to the main text, as mentioned in the reply to the next comment.

The revised paragraph will read as follows:

"Because of the low number of samples (only four periods), the results of the linear regression analyses between the annual area loss rates and climatic parameters are only indicative. None of the correlations were statistically significant at the 95 % confidence level (Table A2). At Peera, three parameters related to the winter air temperatures had the highest correlation coefficients, and p-values < 0.1. These parameters were MWAT ($R^2 = 0.88$), √FDD ($R^2 = -0.88$), and the number of days with air temperature < -10 °C ($R^2 = -0.87$). At Laassaniemi, the area loss rates had very little correlation with the climatic parameters; the lowest p-values were only around 0.3 for snow cover onset ($R^2 = -0.44$) and snow cover duration ($R^2 = 0.46$)."

29. Why not have a figure similar to Fig 4 for your area loss? If not, then I think you should at least bring Table A2 into your main text, as I think it is more important than Table 3 and Fig 9.

We do not think that a figure showing the relationships between annual area loss rates and climatic parameters is necessary, because the number of samples is very low and the lack of statistically significant correlations. We can move Table A2 into the main text, however.

30. Table A1 – Put that the ALT and RTK data are annual from 2007-2021 in the Table Text.

Done.

31. Table A2 – Put that the area loss data is from 1960-2021.

Done.

32. Line 299- The palsas in Olvmo et al 2020 are also larger than those in your study. Would be good to put the size of the palsas from Olvmo et al in the discussion. As you write, Borge et al (and I think Seppälä too) talks about the importance of the morphology in relation to degradation.

That is a good point. We will rephrase this sentence so that the difference in the palsa area sizes is clear.

33. Line 315 – are they "more important" than climate? Or merely "also important factors"? I think the latter.

This is a very good question. After thorough consideration, we concluded following:

Based on our results, it does seem that other factors are more important than climatic variables in regulation of the seasonal thaw at Laassaniemi. However, more data and comprehensive statistical analysis are needed to confirm this statement. Therefore, we decided to reword Line 315 of the original manuscript from "… are more important factors regulating the seasonal thaw…" to "… have strong effect on the seasonal thaw…"

34. *Also, do you think your use of only "top of palsa" area measurements of ALT has led to a lack of a strong correlation with climatic factors, particularly in the larger of the two palsas you study?

As mentioned above in the reply to the comment #9, we have checked the correlations with the climatic parameters using all ALT values $\leq 1$ m. The results showed less correlation with the temperature-related parameters and better correlation with precipitation-related parameters when all $\leq 1$ m values were used. However, the improvements in the correlations were not high enough to change the p-value from $> 0.05$ to $\leq 0.05$, for any of the parameters at either site. Thus, we conclude that using only TOP-ALT values was not the reason for the lack of significant correlations.

35. Line 351 – rather than say "the permafrost area in 2021 was less than 25% of that in the 1960s" I would say that "the palsas in 2021 have shown a lateral degradation of 75% *(or whatever the number is…)* the 1960 areal coverage", since that is what you really assessed that with the aerial photos. What area exactly the permafrost is (an internal characteristic that you aren't seeing with the images), isn't necessarily the same as the extent of the palsa at the time you image it.

We agree, that palsa extent is not the same as permafrost extent. We will rephrase this sentence as follows: "At the Peera and Laassaniemi palsa mires, **the extent of palsa area degraded over 75 % between 1959 and 2021.**" In addition, we will make sure that we do not use 'permafrost extent' when we mean 'palsa extent' in the revised manuscript.

**Corrections and text improvements**

36. Line(L) 17 – "its extent" is vague. Replace with a better geographical noun – whether " the Arctic" or "the Arctic permafrost region".

By "it's extent" we mean that the permafrost degradation is observed not only in the Arctic, but also in the mountain environments, the Tibetan Plateau, and non-glaciated areas of the Antarctic. We will edit this sentence to include the list of the regions with permafrost.

37. L23 – Write so it is more clear… "The main difference between peat plateaus and palsas are in…"

Done.

38. L31 – mires'

Corrected.

39. L49 – "ALT varies from a …"

Corrected.

40. L146 – "UA system settings" should be "UAS settings"

We will add "the" in front of "settings" to make it more clear that we mean variations in the UA systems and the settings used in the data collection.

41. L86 – palsas'

It is true that the surrounding vegetation complicate delineation of palsas from aerial data and DEMs in general. However, in this case, "palsa's edges" refer to the edges of the Laassaniemi palsa. Therefore, we do not change this as suggested.

42. L132 -aerial

Corrected.

43. Fig 5 should appear before Table 1, according to the earlier reference to it in the text (at Line 212).

Thank you for pointing this out. We will edit the placement of figures and tables to follow the order in which they appear in the text in the revised manuscript. Their final placement in the article depends on the typesetting process, however.

44. L332 – "…in which November …" Delete "the". Or, do you even need this clause?

We will remove the clause about November being included in the winter months.

45. L333 &359 – Arctic (I think it should be capitalized when used as Arctic region)

Corrected.

46. L334 – …ground's thermal…

Corrected.

**References**

Liston, G.E. and Elder, K. A.: Distributed Snow-Evolution Modeling System (SnowModel), J. Hydrometeorol., 7, 1259–1276, https://doi.org/10.1175/JHM548.1, 2006.

Störmer, A.: Modelling snow distribution over discontinuous permafrost related to climate change in Kilpisjärvi, Finnish-Lapland, Störmer, A.: Modelling snow distribution over discontinuous permafrost related to climate change in Kilpisjärvi, Finnish-Lapland, M.S. thesis, Faculty of Natural Sciences, Gottfried Wilhelm Leibniz University of Hannover, Germany, 2020.

Tomhave, L.: Palsa Development and Associated Vegetation in Northern Finland. Störmer, A.: Modelling snow distribution over discontinuous permafrost related to climate change in Kilpisjärvi, Finnish-Lapland, B.S. thesis, Faculty of Natural Sciences, Gottfried Wilhelm Leibniz University of Hannover, Germany, 2018.

---

## Author Comment (AC2)

**Our point-by-point replies (in black) to the question and comments by Reviewer 2 (in blue).**

The manuscript "Permafrost degradation at two monitored palsa mires in north-west Finland" by Verdonen et al. presents an in-depth analysis of palsa degradation at two sites, relying on multi-year field data from a variety of sources. The manuscript is well written and I recommend publication after addressing the following comments:

We thank Reviewer 2 for insightful comments and many good questions, and are grateful for their time and effort in providing valuable feedback. We believe that addressing the issues raised by Reviewer 2 will substantially improve our manuscript, particularly regarding the descriptions of the methods applied.

1. Sect. 3.2: Please provide further details on the DEM generation, i.e. the number of GCP's employed, the accuracy in lateral and vertical direction as provided by the photogrammetry software. Please also provide details on how consistency in time was ensured, i.e. are there any stable points in the DEM's that could be used to check whether there are global offsets between individual years? The authors write "Variations in the UA systems, settings used in the data collection and the devices used to collect the coordinates of the GCPs resulted in discrepancies in the DEMs of different years. Therefore, we used the palsa polygons as delineated from the 2016 orthomosaic to extract only the areas of the main palsas from the DEMs. We then used the minimum value within that area as the base altitude for the respective year." It would be nice to motivate this procedure (which I don't question) from the uncertainties inherent in the DEM generation procedure, at least to some extent.

Thank you for pointing out missing details regarding generation of the DEMs. We agree that these procedures and uncertainties should be described better. Reviewer 1 also asked for more details about the aerial data used in this work. Therefore, we will add more information about the UAS DEM processing and uncertainties due to vegetation cover on the palsas in the first paragraph of Section 3.2, and add a table with details including acquisition dates, spatial and spectral resolutions, number of GCPs, and XYZ errors in the Supplementary materials. We will also change DEMs to DSMs when referring to the UAS-based elevation models as advised by Reviewer 1.

Unfortunately, we do not yet have fixed GCPs, which would allow a more accurate comparison of the different UAS DSMs (as suggested by Fraser et al., 2022). Thus, the only objects that could potentially be used as "stable" reference points at our sites were boulders around the mire areas. The issue with them is that they are located at the edges of the areas covered by the UAS surveys, where the "edge effect" causes uncertainties in the elevation.

2. 162 ff: Please explain in more detail why SnowModel is a suitable tool to reproduce snow dynamics in the extremely challenging environment on top of a palsa, i.e. present some key elements of the model physics, in particular on wind redistribution. The validation provided in favor of the model relies on an unpublished master thesis which I wasn't able to access with a quick Google search. Please provide more details on this work in the manuscript, i.e. include the main findings of the thesis in this study. From the information provided, it is not possible to assess whether the modeled snow data allow for a sounds assessment of long-term trends, thus also

affecting the Results section. Additional validation on snow onset and melt-out could possibly be obtained from remote sensing data, e.g. Sentinel-2, at least for years with infrequent cloud cover in the respective periods.

In addition to the validation presented in the Master's thesis of A. Störmer (2020), we are confident that the SnowModel (Liston and Elder, 2006) is suitable (although not perfect) tool to estimate snow dynamics on palsas because it takes into account different meteorological variables and climatic processes, local topography and vegetation height. In short, the SnowModel is a compilation of four sub models: (1) SnowPack to simulate changes in snow depth based on temperature, precipitation, melting and sublimation processes, (2) SnowTran-3D to simulate the effects of wind, suspension, saltation, snow erosion at slopes, accumulation at the end of hill slopes, and vegetation, (3) MicroMet to simulate the basic distribution of meteorological input for the investigation area, and (4) EnBal to simulate the energy balance of the surface based on the meteorological inputs of the MicroMet model.

We agree that the description of the SnowModel is too short in the current version of the manuscript. We did not describe it in more detail, as it was a complementary addition to the analysis to check whether the local snow depth values would show better correlation with the ALT compared to the values measured at the meteorological station in Kilpisjärvi. We also agree that the unavailability of the thesis, which is referred to, makes it impossible for the reader to check further details. Thank you for pointing this out! To address this issue, we will include a more detailed description of the SnowModel, the key elements of the model physics, the parameters used, potential issues, and a summary of the results from A. Störmer's thesis, which are relevant to the presented work, in the Supplementary materials of the revised manuscript.

Validation of the snow-on and snow-off dates with freely available optical satellite data, such as Sentinel-2 or even Landsat, would be a good addition. Unfortunately, frequent cloud cover over the study areas resulted in very few images that could be compared with the dates derived from the snow depth data at the Kilpisjärvi Weather Station. The few available images for the snow-off dates confirm what could be also expected based on the lower accumulation of snow on the palsa surface and game camera recordings at Peera from autumn 2016 until end of summer 2017, the actual snow-off dates are generally earlier for palsas than observed at the weather station. We will include this point in the description of the snow parameters in Section 3.3. Fewer images are available for the snow-on dates at our sites, but based on the game camera images, the difference is much less pronounced.

3. Sect. 4.1 The negative trend for the second site is very interesting – please add 1-2 sentences to highlight the procedure again, in particular that only values from the TOParea, i.e. the still stable part in later years, are compared. It is easy to miss this as a casual reader.

We will add clarifications that the values in this case are top-of-palsa ALTs. We will also include the results using all ALT values $\leq$ 1m to evaluate the effects of using only TOP values on temporal trends and correlations with climatic parameters. Because of this addition, we will state clearly throughout the manuscript whenever we refer to the top-of-palsa ALT.

4. Fig. 4: It is not really clear to which site the regression parameters and the R2 values belong, the one on the top also has a different color in some of the plots?

We agree that the colours of the texts within Figure 4 are confusing. We will edit the colours of the equations and $R^2$-values so that the ones for Peera will be in black, and the ones for Laassaniemi will be in light blue similar to the point symbols.

5. Sect. 4.2 I don't think it makes sense to present correlations that are not statistically significant, even if there is a trend. This is exactly the point of a statistical significance test. So for me the main conclusion of this section is that ALT is not strongly controlled by any of the tested parameters, except for the ones pointed out by the authors as significant. But also for these, it would be good to discuss the level of significance some more. This in itself is a very important result, in particular that the clear decrease in ALT for the second site does not seem to be controlled by larger-scale climatic drivers, but more by local factors which the authors cannot quantify at this point. I do not question the analysis (except perhaps the snow data, see above), but I think this section needs to be rewritten to some extent.

We kindly disagree about the relevance of including non-significant correlations, such as relationships between the ALT and air temperatures (Fig. 4 a, b, d, and e) in this case. The air temperatures have been found to be important factors controlling the ALT in palsas in other studies (Åkerman and Johansson, 2008; Sannel et al., 2016; Mamet et al., 2017), which makes it interesting that the relationships with these parameters were not that strong at our sites. And that is why we think it is important to present them in the results. We will also include this point in the discussion, as it has not been addressed in the submitted manuscript.

In the revised version, we will also include the $R^2$-values in the main text, as advised by Reviewer 1, and rewrite Section 4.2, so that it states more clearly, which correlations were statistically significant.

6. Table 2: can you add the corresponding data, i.e. 2016 and 2021, from the dGPS surveys to this table! The difference in absolute values seems to be significant between the two methods, so having a direct comparison of the same time slices is important.

The UAS DSMs, which were used to derive the values presented in Table 2, cover larger areas than the RTK-GNSS surveys from the same years. For this reason, we do not include the RTK-GNSS DTM values for 2016 and 2021 in the same table. Instead, we decided to add a new Section 4.5, where we will compare palsa height changes 2016–2021 based on the two methods. In this new section, we will also include the RMSEs for the elevation and height values between the RTK-GNSS and UAS DSMs.

7. Fig. 6: why are there two color legends (one in meter, the other in cm)? I think it could also be good to adjust the color scales and not use confining max-min-values. Right now one mainly sees the areas of full collapse, but it is equally important to be able to assess to what extent the main areas of the palsa have subsided. Furthermore, the authors should clarify to what extent the increases near the palsa edge are due to vegetation (i.e. is there vegetation of such height at all? Is the first survey taken after leaf-fall and the second before?), or the result of consistency issues between the DEM's, like global shifts, tilts or rotations (see also comment on DEM accuracy above).

We will enlarge the legend text and rearrange Figure 6 so that it is more clear that one legend refers to the differences in metres and the other refers to the differences in per cent. We wanted the same colours to indicate the same values for both palsas. To achieve this, we used 25 classes with 10 cm intervals (except for the min and max classes). The continuous legend for the changes in metres is a compromise, as showing all the 25 classes in the figure legend does not seem reasonable to us, and using fewer classes hinders changes at Laassaniemi. We acknowledge that it is difficult to assess height changes in the core areas of the palsas, however. For the revised version, we will look more into the options and will either include contours or adjust the colours to address this issue.

The leaves were still on during the UAS surveys in 2016 and 2021. The vegetation height varies from a few centimetres to ca. 50–60 cm. The shrub cover is particularly high at the northern edge of the Peera palsa. Although some displacement of the vegetation may have occurred, the apparent increases at Peera are more likely due to differences in the quality of the UAS data and the resulting DSMs. This will be clarified in the main text of Section 4.4 and in the caption of Figure 6. At the northern edge of the Laassaniemi palsa, the change was indeed due to vegetation growth, as the common cotton grass has expanded rapidly within the thermokarst pond.

8. 259ff: I am not sure about these correlations, is there any statistical significance? Also, the authors write that a higher value of snow onset (=later snowfall) correlates with a higher degradation rate for the second site (true?), but there is no correlation to e.g. fall air temperature? A later snow onset should rather lead to more ground cooling, except when the air temperatures are above freezing. If the data are like that, it is important to state this result, but the authors should check to what extent such correlations are statistically supported.

We will rephrase the paragraph about the correlations between the annual area loss rates and the climatic parameters, so that it states more clearly the lack of significant correlations and that the regression results are rather indicative because of the low number of samples. The $R^2$-values of the correlations mentioned in the text will also be included. In addition, we will move Table A2 into the main text, as advised by Reviewer 1.

The revised paragraph will read as follows:

"Because of the low number of samples (only four periods), the results of the linear regression analyses between the annual area loss rates and climatic parameters are only indicative. None of the correlations were statistically significant at the 95 % confidence level (Table A2). At Peera, three parameters related to the winter air temperatures had the highest correlation coefficients, and p-values < 0.1. These parameters were MWAT ($R^2 = 0.88$), $\sqrt{FDD}$ ($R^2 = -0.88$), and the number of days with air temperature < -10 °C ($R^2 = -0.87$). At Laassaniemi, the area loss rates had very little correlation with the climatic parameters; the lowest p-values were only around 0.3 for snow cover onset ($R^2 = -0.44$) and snow cover duration ($R^2 = 0.46$)."

Indeed, the regression results in Table A2 imply that at Peera, the area loss rates are higher in periods with later snow onset, although the correlation is not significant. At Laassaniemi, this relationship is opposite and is in line with the negative correlation between the ALT and snow onset at both sites (thinner ALT after later snow onset in the preceding winter). The opposing relationships are in agreement with our interpretation of the results concluding that the palsa area loss at Peera is more related to the changes in air temperatures, and at Laassaniemi, the area loss is more related to the changes in snow cover.

We did not discuss the correlation with autumn air temperatures. However, quick analysis showed that the area loss rates did not correlate with autumn (Sept., Oct., Nov.) air temperatures at either site. The analysis showed also that the mean autumn air temperatures have increased significantly 1960–2021 (+ 0.3 C per decade, $R^2 = 0.1$, p = 0.012), and the snow cover onset correlates with autumn temperatures ($R^2 = 0.35$, p < 0.001).

**References**

Fraser, R., Leblanc, S., Prevost, C., and van der Sluijs, J.: Towards Precise Drone-based Measurement of Elevation Change in Permafrost Terrain Experiencing Thaw and Thermokarst, Drone Syst. Appl., https://doi.org/10.1139/dsa-2022-0036, 2022.

Liston, G.E. and Elder, K. A.: Distributed Snow-Evolution Modeling System (SnowModel), J. Hydrometeorol., 7, 1259–1276, https://doi.org/10.1175/JHM548.1, 2006.

Mamet, S., Chun, K.P., Kershaw, G.G.L., Loranty, M.M., and Kershaw, P.: Recent Increases in Permafrost Thaw Rates and Areal Loss of Palsas in the Western Northwest Territories, Canada, Permafrost Periglac., 28, 619–633, https://doi.org/10.1002/ppp.1951, 2017.

Sannel, A.B.K., Hugelius, G., Jansson, P., and Kuhry, P.: Permafrost Warming in Subarctic Peatland – Which Meteorological Controls are Most Important? Permafrost Periglac., 27, 177–188, https://doi.org/10.1002/ppp.1862, 2016.

Störmer, A.: Modelling snow distribution over discontinuous permafrost related to climate change in Kilpisjärvi, Finnish-Lapland, M.S. thesis, Faculty of Natural Sciences, Gottfried Wilhelm Leibniz University of Hannover, Germany, 2020.

Åkerman, H.J. and Johansson, M.: Thawing Permafrost and Thicker Active Layers in Sub-arctic Sweden, Permafrost Periglac., 19, 279–292, https://doi.org/10.1002/ppp.626, 2008.

---

## Author Response (AR1)

**Authors' response**

Below are our point-by-point responses (in black) to each of the comments (in blue). All page and line numbers in our replies refer to the revised manuscript pdf file with tracked changes. Added or rephrased texts are italicised.

In addition to the revisions mentioned in our replies, we did many small changes throughout the manuscript. The most significant change is that we corrected 2007–2021 RTK-GNSS measurements from palsas based on more accurate and precise coordinates of the geodetic control points. This caused small changes in the base altitudes that were used to derive palsa heights from the RTK-GNSS DTMs (P 5: L 129, P 15: L 294, and footnotes of Table 1). As the correction affected also XY-coordinates of the RTK-GNSS points at Laassaniemi, we incorporated this shift in Figures 2c and 2d, and in Figure 6e (Fig. 6d in the old manuscript version).

**Comment from the editor:**

One quick from my side is that there are a couple of cases where the reviewers ask about correlations, where the authors say they have investigated but did not include in the text/figure because the results did not come out significant or no correlation was shown. I think it is worth mentioning this briefly in the main text for those readers who might wonder the same things as the reviewers.

Thank you for this suggestion. We clarified this in the main text at the beginning of Section 4.2 (P 11, L 237–239) as follows:

"*Most of the correlations between climatic parameters and the $ALT_{TOP}$ were not significant. Therefore, we focused here only on those correlations, which are comparable with other studies, and/or which showed relatively strong correlations at least at one of the sites.* The *full list of* $R2$ and *p*-values of the regression analyses is presented in Table A1.*"

As the comparable correlations in other studies were not mentioned in the earlier version of the manuscript, we added following sentence in the introduction (P 3, L 61–62):

"Some active layer data are also available from palsas and/or peat plateaus (Åkerman and Johansson, 2008; Sannel et al., 2016; Mamet et al., 2017). *Based on these studies, summer air temperatures and thaw degree days, and in some cases freezing degree days and snow accumulation, seem to be the most important meteorological predictors of the ALT.*"

We also incorporated this in the revised version of the paragraph, where we present the results of the linear regressions between the annual are loss rates and climatic parameters on P 22, L 382–383. This paragraph begins with:

"*The full list of R2 and p-values of the correlations between the annual area loss rates and climatic parameters is presented in Table 6.*"

**Comments from the reviewer 1 (Heather Reese):**

The paper by Verdonen et al., is a study on degradation of palsas and permafrost plateaus over two sites in northwestern Finland, where Active Layer Thickness and ground elevation has been measured systematically and annually since 2007, and aerial photos exist from 1960 and onwards (including UAS images) to measure lateral degradation. Verdonen et al user linear regression to relate changes in palsa area and height changes to climatic parameters. This paper contributes to a better understanding of the changes in the sensitive palsa mires of Fennoscandia, and their association with climate variables. It's well written, and represents a further step in understanding palsa dynamics and associated influences, both climatic and otherwise.

We thank Heather Reese (Reviewer 1) for insightful comments and many good questions and are grateful for her time and effort in providing valuable feedback.

Main issues are marked with a *.

**Questions and comments**

1. Line 4 – It would be informative with a short form of the Latitude after the names.

We added "*68° N*" after the site names as suggested on Line 4.

2. Line 5 – I don't think that is true that your study focuses on the time period covered by your UAS data. It focuses just as much on your ALT measurements from 2007-2021 (albeit only within the non-degraded area of the palsas), as well as your RTK-GNSS ground elevation measurements, which to me is the more interesting data set since it should be more accurate. Even the next line in the abstract mentions using the longer time series of aerial photos. So I would take away this sentence ("The emphasis is on detailed change detection …").

Our initial idea was to focus more on detailed change detection during the last 6 years of the investigated period. However, the results of the ALT and RTK-GNSS measurements were more interesting and shifted the focus of this article. Therefore, we agreed with this comment and rephrased the beginning of the abstract (P 1, L 3–7) so that there is no emphasis on the UAS data, and added a sentence, that the analyses were done using linear regression:

"*Here, we present the results of the aerial photography time series analysis (195960–2021), annual RTK-GNSS and active layer monitoring (2007–2021), and annual Unoccupied Aerial System surveys (2016–2021) at two palsa sites (Peera and Laassaniemi, 68° N) located in north-west Finland. We analysed temporal trends of palsa degradation and their relation to climate using linear regression.*"

3. Line 8 – Mention that the ALT data are annual from 2007-2021.

Done (Line 10)

4. L61 – name the years that your study is looking at, rather than using the more vague "the investigation period" phrasing.

We rephrased the first research question (P 3, L 66–67) to include the periods as follows:

*"(1) How did the lateral extent of palsas (1959–2021), palsas' height (2007–2021) and the active layer thickness (2007–2021) change during the investigation periods?"*

5. L66 – It would be better if you indicated what kind of sensor you are using for data collection from the UAS platform. Is it an RGB camera, an NIR camera, and/or a Lidar sensor?

Thank you for the suggestion. Indeed, many different sensors have been used to collect data with drones at these study sites. However, here we used only the data collected using RGB-sensors. We edited the text (P 3, L74) to mention that used sensors were RGB as follows:

"For the last six years (2016–2021), the UAS *RGB* data have enabled capturing changes in palsas at ultra-high *(< 5 cm) spatial* resolution."

6. L91 – Describe the area of the two palsas in the same way and give their dimensions, as the shape of the palsa may affect how it reacts.

We revised the descriptions of the two palsas so that their dimensions and particularly differences in shape and size are more clear. In the revised version, the dimensions of the Peera palsa are described as follows (P 4–5, L 99–101):

*"The palsa consists of a main body with a diameter of around 50 m, the highest point of which was 2 m above the surrounding mire in 2021, and two lower parts, which increase the palsa's core-to-edge ratio. The area of palsa was around 1500 m$^2$ in 2021."*

And the dimensions of the Laassaniemi palsa are described as follows (P 5, L 106–108):

"The detailed investigation focuses on the southern palsa (hereafter 'Laassaniemi palsa'), which *has over ten times smaller area (ca. 120 m$^2$) than the Peera palsa. The Laassaniemi palsa is more oval-shaped and it was around* 18 m long and 7 m wide in 2021."

7. Figure 1 d and f – If these are 1 or 2 m high, are these palsas or peat plateaus? Or were these taller some decades ago? Just double checking, seeing as you made a point about the difference between the two.

We do not have information about the height of Peera palsa before the beginning of the monitoring in 2007, when its highest point was around 2.5 m above the surrounding mire surface. In Laassaniemi, the height of at least one of the palsa mounds was closer to 2 m based on a photograph taken in 1997, indicating that the whole palsa or peat plateau used to be higher than the ca. 1.5 m observed in 2007. Overall, it seems that all palsas in our study areas are disintegrated parts of larger peat plateaus.

We added this information in the site descriptions on P 4, L 98:

"The studied mire area is ca. 5.2 ha and has four palsa mounds, *which used to be two large peat plateaus.*"

and on P 5, L 106:

"The mire includes four palsas, *which are disintegrated parts of a larger peat plateau, and* all of which are < 1 m high (Fig. 1e)."

8.  *In general – I think it would be better if you used the terms Digital Terrain Models (DTMs) and Digital Surface Models (DSMs) instead of the umbrella term DEM, particularly since your article refers to both kinds of elevation models. Or at least conscious use of the terms. Your RTK-GNSS created DTMs while your drone images will create DSMs.

Thank you for pointing this out. Our reasoning to use only the umbrella term "DEM" was to avoid confusion by using too many similar terms. We acknowledge that the result was the opposite, and it is better to state clearly whether we refer to DTMs or DSMs. Therefore, we changed the RTK GNSS-based DEMs to DTMs and UAS DEMs to UAS DSMs throughout the text. We also added information about UAS-based DEM processing in the first paragraph of Section 3.2 (P 6, L 156–160) as follows:

"The data were processed into orthomosaics and DEMs in Agisoft Metashape software (version 1.8.4), *by using the structure from motion approach (Westoby et al., 2012). Aggressive depth filtering was applied when processing the UAS data into a dense point cloud to minimise the effects of small shrubs or shrub patches. In some parts of the study areas the vegetation cover was too dense to distinguish the ground surface from the point cloud. Therefore, the DEMs created are Digital Surface Models (DSMs).*"

9.  *L120 – Do you mean that if there was lateral degradation in any year from 2007-2021 that you did not include this in the ALT measurements used in the regressions? If so, that should have some effect on your result (and maybe this is why you don't see strong relationships between ALT and climate parameters at the larger of the two palsas). Can you motivate your choice and make clear how using only the "Top of Palsa" mean ALT measurement can affect your results in the Discussion section. It seems like you would be missing the bigger changes. You can see the points you are missing when looking at Fig 2.

This is a very good point, and we agree that the effect of using only top-of-palsa (TOP) ALT values could have been motivated and discussed more in the manuscript. Only the values within the delineated TOP-areas were included in the calculation of the mean ALT and regression analyses. Our assumption was that we could better capture the relationship(s) between climatic parameters and annual variations in the ALT by excluding the areas most affected by lateral thermal fluxes. Preliminary analyses of all ALT values ≤ 1 m also showed very similar trends to the trends using only the TOP-values, which further justified focusing on only the most "stable" parts of the palsas.

To avoid speculations in the discussion regarding the effects of using only the TOP values, we ran the same regression analyses with the mean ALT values calculated from all ≤ 1 m measurements and included these analyses in the manuscript text and in the appendices (Fig. A1 and new Table A2). We rewrote the paragraph (P 6, L 133–140), where we motivate our use of TOP ALT.

The old text was following:

"We delineated the areas not visibly affected by lateral degradation of permafrost to analyse the effects of climatic drivers on the active layer without the effect of thermokarst at the edges of the palsas (top-of-palsa, Fig. 2). The ALT refers to this top-of-palsa ALT in the rest of this article unless indicated otherwise. The trends in the ALT were analysed by fitting linear regression models using the least-squares method."

Revised version is following:

*"The ALT trends were analysed by fitting linear regression models using the least-squares method. Only ALT values ≤ 1 m were included in the statistical analyses. To capture seasonal thaw dynamics with minimal effects of lateral thermal fluxes and thermokarst at the edges of the palsas, we delineated the areas not visibly affected by lateral permafrost degradation. These top-of-palsa (TOP) areas were 309 m² at Peera and 49 m² at Laassaniemi (Fig. 2). The focus of our analyses and results is on the ALT values within these TOP-areas (ALT$_{TOP}$), while the results using all ALT values ≤ 1 m are presented in the Appendices (Fig. A1 and Table A2). The number of ALT values ≤ 1 m (all and within TOP) for each year are presented in Table S1."*

We also added short descriptions of the results using all ALT values ≤ 1 m in the Results on P 9, L 225–226:

*"Linear trends 2007–2021 using all ALT values ≤ 1m show similar results (+ 0.4 cm a$^{-1}$ at Peera, - 1 cm a$^{-1}$ at Laassaniemi), although with higher standard deviations (Fig. A1, Table S2)."*

And P 13, L 263–267:

*"Using all ALT values ≤ 1m instead of ALT$_{TOP}$ in the regression analyses slightly improved the correlations with some of the precipitation-related parameters at both sites (Table A2). The improvements in the correlations were not high enough to change the p-value from > 0.05 to ≤ 0.05, for any of the parameters at either site, however. The p-value increased over the 0.05 threshold for the number of days with air temperature > 15 °C and for rainfall during September and October of the previous year for Peera, and for snow cover onset at Laassaniemi when all ALT values ≤ 1m were used compared to ALT$_{TOP}$."*

10. Section 3.2 – More details are needed on the sensors and specifications used to create these data. A Table could be useful here. What camera? What scale (or GSD- ground sampling distance) are the original images taken at? What full date? Which photo dates were the panchromatic, and what were the others? With the UAS, what platform (since this helps indicate which GPS was used)?

We added as supplementary materials Table S2 indicating the details of the aerial data from the National Land Survey of Finland, and Table S3 indicating details about UAS data used in this work

While looking for more details about the aerial photographs we found a mistake in the acquisition year of the first photographs. The correct year is 1959, as opposed to 1960, which was indicated in the web platform where we accessed the data. We have corrected this throughout the manuscript and in the calculations of the annual area loss rates and the results of the regression analyses shown in Table 6 (Table A2 in the original manuscript version) on P 25–26.

11. *L145 – You listed a number of issues that you ran into. In addition to this, the UASbased data result in elevations that include vegetation heights (DSMs) and are therefore not completely reliable for showing accurate elevation from year to year, and therefore subsidence and volume changes over time. How tall is the vegetation on the palsas? In any case, this should be a primary reason why you can't calculate reliable volume changes from these data. I would reword this section so that this is acknowledged. However, the orthophotos are useful. It will be much better when you get the UAS-Lidar data for calculating volume changes!

Vegetation height varies from zero centimetres (bare peat surface) to ca. 50 cm. We agree that the effect of vegetation cover on the UAS-based heights and volumes should be addressed more in the manuscript. Therefore, in the revised version, we added the uncertainties due to the vegetation cover

along with the issues regarding georeferencing (see our reply to the next comment) to the description of the UAS DSMs processing (P 7, L 172–176):

"*Change analysis based on UAS DSMs is sensitive to small changes in vegetation structure and differences in lightning conditions. No notable changes in the vegetation cover of the palsas were observed during the annual visits to the sites by the authors. However, it is possible that small differences in vegetation cover because of, for example, trampling or vegetation growth may have affected the height and volume changes derived from the DSMs.*"

12. L145 – I think you should also indicate how you geo-referenced your UAS data. You mention problems with the equipment.

Since you have RTK-GNSS data taken annually, couldn't you calculate an RMSE for elevation of the UAS DSMs, indicating their potential error? Then again, that would mean you are comparing DTM and DSM. But still, you might be able to observe systematic errors across the UAS DSM. When you mention in results that you see a trend from southwest to northeast (Line 225), I wonder if it is due to a tilting of the UAS DSM, which can easily happen when good georeferencing isn't possible.

My main point here is not to re-do a lot of work or invest a lot more time in the UAS DSMs, because frankly these will always include uncertainty due to 1) including all surface heights and 2) poor geolocation accuracy if not fixed with RTK-GPS control points. I think you just have to admit and realize the weaknesses of that data set for accurately measuring subsidence.

Reviewer 2 also raised the question about the georeferencing and processing of the UAS DSMs. In addition to the lack of consistency regarding the equipment and UAS data collection, the comparison of the UAS DSMs is further complicated by the lack of stable and fixed control points within the study areas. We clarified this in the revised manuscript (P 7, L 177–178):

"*Due to the lack of stable surfaces unaffected by the "edge-effect" in the UAS data, we were unable to correct for potential global offsets between the DSMs.*"

As recommended, we calculated the RMSEs between the RTK-GNSS elevation values and the UAS DSM values for 2016 and 2021. They are presented in the new Table 4 on page 19.

We have checked the DSMs of the Laassaniemi palsa for potential tilting, as suggested, and compared them to the RTK-GNSS-based DTMs from the same years. The subsidence in the Laassaniemi DTMs is also highest in the northeastern corner and lowest at the southwestern end. Therefore, we ruled out a tilting effect. However, in 2021, the surface of the Laassaniemi palsa was surprisingly higher based on the RTK-GNSS DTM than derived from UAS DSM. This difference and further comparison of the palsa changes based on the RTK-GNSS DTMs and UAS DSMs are now presented in new Section 4.5 (P 18–19, L 335–353) of the revised manuscript where we compare the changes in the palsas based on the UAS DSMs and RTK-GNSS DTMs.

We also added a paragraph in Section 3.2 (P 7, L 181–185) to describe, how we compared the 2016 and 2021 RTK-GNSS DTMs and UAS DSMs:

"*We used two methods to assess the results of height changes based on the 2016 and 2021 UAS DSMs. First, we used TOP polygons to extract the mean 2016 and 2021 heights from the RTK GNSS DTMs and UAS DSMs and compared these mean height values. Second, we used all RTK-GNSS*

*measurement points to extract the absolute elevation (m a.s.l.) and height values to these points from the UAS DSMs and RTK-GNSS DTM heights. We then calculated root mean square errors (RMSEs) and median differences between the elevation values and height values of the two datasets.*"

13. L165 – Again this is a DSM and not a DTM (or DEM), with vegetation included. Finland has a national Lidar scanning – why didn't you use the DTM from that for the snow model (or even better, both)? Too coarse? Can this account for the rather large differences between modelled and the reference snow depth measurements (10-30cm difference)? Also, where was the vegetation classification from? Your own? In any case, what classes were there?

Yes. The 2 m DEM (or DTM) from the National Land Survey of Finland is too coarse in this case, as the idea was to model the snow distribution with considering small-scale variations in the topography.

Vegetation classification of the Peera palsa mire used by A. Störmer (2020) in his Master's thesis was performed by another student (Tomhave, 2018), who mapped the vegetation using UAS orthomosaics and whose results were validated in the field. This classification was then adapted into the classes used in the SnowModel, in which there are predefined vegetation types with associated snow-holding capacities (see Table 1 in Liston and Elder, 2006). Following SnowModel vegetation types were used with the classes from the classification by Tomhave (2018) in the brackets:
- Erect Shrub Tundra (Dwarf Birch)
- Low Shrub Tundra (Dwarf Shrub)
- Prostrate Shrub Tundra (Lichen)
- Arctic Gram, Wetland (Sphagnum, Peatland vegetation)
- Bare (Bare rock, Peat)
- Water (Water body)

These same classes were also used to classify vegetation types in the Laassaniemi palsa mire by A. Störmer for the use in this work.

To further address this comment and another reviewer's questions regarding the SnowModel, we included a more detailed description of the SnowModel and its application to estimate snow depths on the palsas in the Supplement.

14. *L170 – I don't think the explanatory parameters are clearly given. A table could help here, or else you could more clearly state it in the text. For example, did you not test any precipitation variable, besides snow?

The explanatory parameters are introduced in the second paragraph of Section 3.3, where we briefly mentioned that precipitation was calculated for different seasons. We have now added more information about the parameters related to precipitation on P 8, L 192–193:

"…and precipitation*: total precipitation, rain (precipitation when air temperature > 0 °C), snow (precipitation when air temperature < 0 °C), and rain during preceding autumn, spring and summer.*"

We also added a reference to Tables 6 (old Table A2) and A1, where full lists of parameters used are given (P 8, L 192–193):

"*Full lists of the climatic parameters used in the analyses are presented in Tables 6 and A1.*"

15. **15.** Fig 2 – I found it hard to see the outline of the palsa. Maybe a little thicker. Also, you should mention what your image is in the background of the 2021 images, and what date it was taken.

We increased the line thickness of the palsa outlines and added dates of the UAS orthomosaics used as the background in Figure 2 as suggested.

**16.** Fig 3 – Very nice information! This figure raises a lot of questions for me, such as What happened in 2012-2014 to cause this change in ALT?. Also, why the divergence in responses between the two palsa sites after 2014? I interpret the large error bars on Peera to indicate the faster degradation in process, likely due to the small size of the palsa, and the high edge-to-core ratio. Do you think the 2014 ALT measurement is correct for Peera? What causes it to be the biggest thaw measurement in case it is correct?

Thank you for these thought-provoking questions. We checked the data again. Almost all 2014 ALT values were higher at Peera than in 2013, despite the earlier measurement dates in 2014. Therefore, we are confident that there are no errors resulting in higher ALT values compared to other years. The top-of-palsa area is 309 m$^2$ at Peera and 49 m$^2$ at Laassaniemi. Larger area and more variability in the vegetation cover allows larger spatial variability in the ALT at Peera, which could explain longer "whiskers" in Figure 3. Based on the regression analyses, the mix of warm summers in 2013 and 2014 and early snow onset in the autumn of 2013 could be the reason for the deeper thaw in 2014. In addition, based on the regression analyses, the ALT at Laassaniemi is less sensitive to the mean summer air temperature or the number of particularly warm (> +15 °C) days, which could explain the lack of similar deep thaw in 2014. After 2014, the trends in the TOP ALTs of the two sites were rather similar (- 1.5 cm per year at Peera and -1.8 cm per year at Laassaniemi); thus, the divergence seems to have been caused by this much deeper thaw in 2014 at Peera than at Laassaniemi.

In the revised manuscript, we added information about the size of the TOP-areas on P 6, L 137:

"*These top-of-palsa (TOP) areas were 309 m$^2$ at Peera and 49 m$^2$ at Laassaniemi (Fig. 2).*"

We also added Table S1 in the supplementary materials showing the measurement dates, mean ALT ± 1 SD, and number of observations (all ≤ 1m values and within TOP).

**17.** Line 191 – Give the R2 value of the few mentioned correlated variables in the text.

We added the R$^2$ values of the mentioned variables throughout the Section 4.2. We also revised the section substantially so that it states more clearly, which correlations were statistically significant.

**18.** Line 212 – I find this paragraph to be confusing due to the mix of observing what I interpret you to mean lateral degradation as well as subsidence. It would be good to be clear here. The heading is about subsidence or volume change with the RTK-GNSS and the top of the palsa measurements. Otherwise, did you use RTK-GNSS to map the area loss (lateral degradation)? It is unclear, due to the heading, and then the mix of different vaguely worded "degradations".

We edited Section 4.3 heading to be more clear:

"*4.3 RTK-GNSS-based palsa area, height, and volume changes*"

and added a following clarification in the first sentence of Section 4.3 on P 14, L 283: "Degradation of permafrost *in the form of palsa area and height loss* is noticeable at both palsas... ".

We also added a following sentence on L 286:

"*The volume decreased by 55 % within the area covered by the RTK-GNSS surveys between 2007 and 2021.*"

and removed the sentence about height changes within top-of-palsa areas from L 275–276 so that it is clearer that we analysed height, area and volume within the areas covered by the RTK-GNSS surveys, not only top-of-palsa.

19. A thought: Since you have measurements in both places, what is the relationship between the RTK-GNSS measured annual subsidence and annual change in ALT? You wouldn't expect (intuitively) to see a fluctuation in ALT at the same time as you have a constant loss of palsa height. Would be a very good figure to include, since you have the data. (OK, I see in the Discussion you mention this, and try to explain it).

A comparison of the annual changes in the top-of-palsa ALT and annual RTK-GNSS-based subsidence did not reveal any correlation between the two variables. In other words, the active layer can be thinner or thicker compared to previous year, regardless of the degree of subsidence. The relationship between subsidence and ALT, on the other hand, is more apparent, especially at Laassaniemi, where subsidence has slowed since 2012. At Peera, the temporal trend in subsidence is the opposite. Further analysis of the annual subsidence is beyond the scope of our article in its current version. However, we added Table S5 showing the annual subsidence within the TOP areas, and Figure S4 illustrating the total thaw subsidence since 2007 and the annual variations in the ALT. We also added a reference to them on P 14, L 288–289:

"*The annual subsidence increased at Peera and decreased at Laassaniemi over the last ca. 8-10 years of the monitoring period (Table S5 and Fig. S4).*"

20. Table 1 indicates that your volume change measurements using your DTM from RTK-GNSS is based only on the "Top of Palsa" area. Good to make sure that is clearly stated in the methods.

This was not our intention. Therefore, to avoid confusion, we edited Table 1 caption (P 15, L 295–296) as follows:

"Palsa *mean* height, area, and volume changes in 2007–2021, *from the whole RTK-GNSS measurement area of the years in question*. Note that the RTK-GNSS surveys cover only the western half of the Peera palsa."

21. Line 220 – Include in the sentence that this is height change measured by the UAS DSMs. Also, are you measuring only the "Top of Palsa" area, or what area are you using? To try to figure that out, I read back in methods, where it sounds like you have used the 2016 extent of the palsa, as delineated from the very detailed orthomosaic, so it will be I guess, a different area than "Top of Palsa". Do I interpret that correctly?

Yes. The areas covered by the UAS DSMs differ from the areas covered by the RTK-GNSS DTMs. Unlike the RTK-GNSS surveys, the UAS DSMs cover the investigated palsas completely, which provides better overview of the changes in the palsas, especially at Peera.

We added "*Based on the UAS DSMs, ...*" at the beginning of the section on P 15, L 299, as suggested.

We also edited Section 4.4 heading to be clearer and more concise with the heading of Section 4.3:

"4.4 Area, height, and volume changes based on UAS DSMs"

22. A thought: you would be able to confirm whether subsidence of 20 cm between 2016-2021 found using UAS DSMs corresponds with the subsidence measured by RTKGNSS from the same time period 2016-2021, given that you looked at the same area.

We did consider adding the comparison of palsa changes based on the RTK-GNSS DTMs and UAS DSMs early in the process of preparing the manuscript but left it out from the final version. Based on this comment and the comments of Reviewer 2, such comparison should be included. Therefore, we added new Section 4.5, as mentioned earlier in the reply to the comment #12, showing the differences in the mean top-of-palsa heights derived for 2016 and 2021. In this section, we also included the RMSE for the elevation and height values between the RTK-GNSS and UAS DSMs.

With this comparison, however, we cannot confirm the 20 cm subsidence found in the UAS DSMs as they cover larger areas than the RTK-GNSS surveys (as mentioned in the previous reply). Within the TOP areas and within the overlapping areas, the comparison shows that the mean height change of the UAS DSMs is double that of the RTK-GNSS DTMs. We added this point in Section 4.5 (P 18, L 338–340:

"*The 2021 UAS DSM resulted in lower heights than the RTK-GNSS DTMs at both sites. Therefore, the TOP subsidence is 17 cm (54 %) larger at Peera and 13 cm (48 %) larger at Laassaniemi based on the UAS data compared to the RTK-GNSS.*"

and in the discussion on P 26–27, L 428–433:

"*The UAS-based change analysis also showed ca. 50 % larger changes in the mean heights of the top-of-palsa areas between 2016 and 2021 than the RTK-GNSS based mean heights. The difference is most likely a mixed result of the effect of vegetation cover in the UAS DSMs and the uncertainties related to the variations in the UA systems, the settings used, and the number and quality of the GCP points. This underlines potential issues related to the use of UAS RGB-data for the elevation change detection in the environments affected by thaw subsidence as well as changes in the surrounding mire surface depending on the water level. Adding fixed GCP points (Fraser et al., 2022) to our sites could improve interannual comparability of the UAS DSMs in the future.*"

23. Line 229: Well, you can't measure the internal permafrost with the RGB images which only show the surficial extent of the palsa. Also Line 294 you refer to how UAS data can lead to overestimation of permafrost. The aerial photos, or any surficial representation of the palsa is only that – the representation of the geomorphological form of the palsa. To find the permafrost, which is an internal characteristic, so far the ALT measurements are needed. Also in Line 294 – it wouldn't be only UAS, but also any aerial photo, or even Lidar that would "overestimate permafrost".

That is true. Our intention was to highlight the difference in the extent of the Laassaniemi palsa derived by the two methods and explain the reason for this difference. We rephrased this paragraph so that the emphasis is on the fact that we used the ALT values to delineate the palsa extent from the RTK-GNSS DTMs but did not use them for palsa delineation from the UAS data. The revised paragraph (P 16, L 311–319) reads as follows:

*"Over 30 % larger area of the Laassaniemi palsa as delineated from UAS DSM (122.2 m$^2$) compared to RTK-GNSS and ALT measurements (82.5 m$^2$) is caused by the difference in how the extent of the palsa is defined using these two methods. In the RTK-GNSS and ALT approach, the information about the active layer affected the delineation of the palsa edge, especially in the areas where palsa edge cannot be distinguished based on the topography alone. For the UAS-based delineation, the ALT-values were not used, and the palsa extent is therefore only based on the information about the surface topography and vegetation cover."*

We also changed "permafrost extent" to "*palsa* extent" in the discussion on P 26, L 406 (L294 in the old version) and "permafrost area" to "*palsa* area" in the captions of Figures 6 (L 334) and 8 (L 366).

24. Line 239/240 – Include in the sentence that this measurement is derived from manual delineation of palsa area from the aerial photos from 1960, …2021.

We added "*Based on the manual delineation of palsa area from the aerial photography,…*" at the beginning of the paragraph on P 19, L 355.

25. Line 240 – that's quite a sad loss of area…

Indeed. At these loss rates we will not have our monitored palsas for much longer.

26. Fig 6 – Legend text is pretty small. Also, I was confused about which legend belonged to which square. Maybe better to make the figure a little bigger, and clearly divide the two sides of absolute and relative change maybe with some lines or column names and Legend heading.

We rearranged Figure 6 so that the absolute and relative changes are now more clearly separated and enlarged the legend text for better readability.

27. Fig 7 – Nice map, I like this a lot. Is there a way to make it larger in the publication?

Thank you! We increased the size of Figure 7 on P 20.

28. Line 259 – Much better description of the results is needed here. What was the R2 of the most correlated climatic variables? Without proper description of the result, it is hard to have any discussion, and hard to compare to other studies (eg Olvmo et al. 2020).

We added the R$^2$ values of the highest correlations and rephrased the paragraph (P 22, L 382–390) so that it states more clearly the lack of significant correlations and that the regression results are rather indicative because of the low number of samples. We also moved Table A2 to the main text, as mentioned in the reply to the next comment.

The revised paragraph reads as follows:

*"The full list of $R^2$ and p-values of the correlations between the annual area loss rates and climatic parameters are presented in Table 6. Because of the low number of samples (only four periods), these results are only indicative. None of the correlations were statistically significant at 95 % confidence level. At Peera, three parameters related to the winter air temperatures had the highest $R^2$ and p-values < 0.1. These parameters were MWAT ($R^2 = 0.88$), √FDD ($R^2 = -0.88$), and the number of days with air temperature < -10 °C ($R^2 = -0.87$). At Laassaniemi, the area loss rates had very little correlation with the climatic parameters; the lowest p-values were only around 0.3 for the snow cover onset ($R^2 = -44$) and snow cover duration ($R^2 = 0.46$)."*

29. Why not have a figure similar to Fig 4 for your area loss? If not, then I think you should at least bring Table A2 into your main text, as I think it is more important than Table 3 and Fig 9.

We do not think that a figure showing the relationships between annual area loss rates and climatic parameters is necessary, because the number of samples is very low and the lack of statistically significant correlations. We did move Table A2 into the main text, however, and it is now Table 6 on P 25–26. We also decided to remove Figure 9 as it was not essential, and consequently also removed references to it on P 22, L 377–379.

30. Table A1 – Put that the ALT and RTK data are annual from 2007-2021 in the Table Text.

We added this information as suggested in the caption of Table A1 (P 30, L 531–536). In addition, we clarified that the volume losses were measured form the whole RTK-GNSS measurement areas.

31. Table A2 – Put that the area loss data is from 1960-2021.

Old Table A2 is now Table 6 in the main text. We added the information in the table caption (P 25, L 406–407) that the climatic parameters are from hydrological years 1961–2021 and the aerial photography time series are from 1959–2021.

32. Line 299- The palsas in Olvmo et al 2020 are also larger than those in your study. Would be good to put the size of the palsas from Olvmo et al in the discussion. As you write, Borge et al (and I think Seppälä too) talks about the importance of the morphology in relation to degradation.

That is a good point. We rephrased this sentence on P 27, L 436–438 as follows:

Old version was:

"The effect of local conditions is also demonstrated by an almost three times slower degradation rate at Vissátvuopmi (Olvmo et al., 2020) than at our sites, although the distance between the areas is only 10–20 km."

New version is:

*"For example, despite only 10–20 km distance from our sites, the degradation rates were almost three times slower at Vissátvuopmi palsa mire (Olvmo et al., 2020), where palsas were considerably larger covering almost 49 ha in 2015."*

33. Line 315 – are they "more important" than climate? Or merely "also important factors"? I think the latter.

This is a very good question. After thorough consideration, we concluded following:

Based on our results, it does seem that other factors are more important than climatic variables in regulation of the seasonal thaw at Laassaniemi. However, more data and comprehensive statistical analysis are needed to confirm this statement. Therefore, we decided to reword this text on P 27, L 461 from "… are more important factors regulating the seasonal thaw…" to "… *have strong influence on* the seasonal thaw…"

34. *Also, do you think your use of only "top of palsa" area measurements of ALT has led to a lack of a strong correlation with climatic factors, particularly in the larger of the two palsas you study?

As mentioned above in the reply to the comment #9, we have checked the correlations with the climatic parameters using all ALT values ≤ 1 m. The results showed less correlation with the temperature-related parameters and better correlation with precipitation-related parameters when all ≤ 1 m values were used. However, the improvements in the correlations were not high enough to change the p-value from > 0.05 to ≤ 0.05, for any of the parameters at either site. Thus, we conclude that using only TOP-ALT values was not the reason for the lack of significant correlations.

We added this point in the discussion on P 27, L 455–459:

"*Excluding the areas affected by lateral thermal fluxes did not obscure any significant relationship between the ALT and climate dynamics. On the other hand, it did result in a few more statistically significant correlations compared to regressions using all ALT values ≤ 1m. For example, at Laassaniemi, none of the regressions showed a significant correlation between the active layer and climatic parameters when all ALT values ≤ 1m were used.*"

35. Line 351 – rather than say "the permafrost area in 2021 was less than 25% of that in the 1960s" I would say that "the palsas in 2021 have shown a lateral degradation of 75% *(or whatever the number is…)* the 1960 areal coverage", since that is what you really assessed that with the aerial photos. What area exactly the permafrost is (an internal characteristic that you aren't seeing with the images), isn't necessarily the same as the extent of the palsa at the time you image it.

We agree, that palsa extent is not the same as permafrost extent. We rephrased this sentence as follows (P 29, L 499–500):

"At the Peera and Laassaniemi palsa mires, the *extent of palsa area degraded over 75 % between 1959 and 2021.*"

**Corrections and text improvements**

36. Line(L) 17 – "its extent" is vague. Replace with a better geographical noun – whether " the Arctic" or "the Arctic permafrost region".

By "it's extent" we mean that the permafrost degradation is observed not only in the Arctic, but also in the mountain environments, central Asia, and non-glaciated areas of the Antarctic. We rephrased this sentence (P 1, L 18–20) to include the list of the regions with permafrost and changed the references to be more specific:

"Permafrost degradation manifested by widespread thermokarst activity *(Kokelj and Jorgenson, 2013),* warming ground temperatures *(Biskaborn et al. 2019)* and (mostly) increasing active layer thicknesses (ALT) *(Smith et al. 2022)* is taking place throughout its extent *in the polar regions, mountain environments and central Asia.*"

37. L23 – Write so it is more clear… "The main difference between peat plateaus and palsas are in…"

Done as suggested on P 2, L 26.

38. L31 – mires'

Corrected on P 2, L 35.

39. L49 – "ALT varies from a …"

Corrected on P 3, L 53.

40. L146 – "UA system settings" should be "UAS settings"

We added a reference to Table S3 right after the "UA systems" on P 7, L 176, which makes it more clear that we mean variations in the UA systems and settings used in the data collection.

41. L86 – palsas'

It is true that the surrounding vegetation complicate delineation of palsas from aerial data and DEMs in general. However, in this case, "palsa's edges" refer to the edges of the Laassaniemi palsa. Therefore, we did not change this as suggested.

42. L132 -aerial

Corrected (P6: L 152)

43. Fig 5 should appear before Table 1, according to the earlier reference to it in the text (at Line 212).

Thank you for pointing this out. We changed the placement of figures and tables to follow the order in which they appear in the text. Their final placement in the article depends on the typesetting process, however.

44. L332 – "…in which November …" Delete "the". Or, do you even need this clause?

We removed the clause about November being included in the winter months on P 28: L 480.

45. L333 &359 – Arctic (I think it should be capitalized when used as Arctic region)

Corrected on P 28, L 481 and P 29, L 508.

46. L334 – …ground's thermal…

Corrected on P 28, L 482.

**Comments from the reviewer 2:**

The manuscript "Permafrost degradation at two monitored palsa mires in north-west Finland" by Verdonen et al. presents an in-depth analysis of palsa degradation at two sites, relying on multi-year field data from a variety of sources. The manuscript is well written and I recommend publication after addressing the following comments:

We thank Reviewer 2 for insightful comments and many good questions and are grateful for their time and effort in providing valuable feedback.

47. Sect. 3.2: Please provide further details on the DEM generation, i.e. the number of GCP's employed, the accuracy in lateral and vertical direction as provided by the photogrammetry software. Please also provide details on how consistency in time was ensured, i.e. are there any stable points in the DEM's that could be used to check whether there are global offsets between individual years? The authors write "Variations in the UA systems, settings used in the data collection and the devices used to collect the coordinates of the GCPs resulted in discrepancies in the DEMs of different years. Therefore, we used the palsa polygons as delineated from the 2016 orthomosaic to extract only the areas of the main palsas from the DEMs. We then used the minimum value within that area as the base altitude for the respective year." It would be nice to motivate this procedure (which I don't question) from the uncertainties inherent in the DEM generation procedure, at least to some extent.

Thank you for pointing out missing details regarding generation of the DEMs. We agree that these procedures and uncertainties should be described better. Reviewer 1 also asked for more details about the aerial data used in this work. Therefore, we added more information about the UAS DEM processing as described in the earlier reply to the comment #8 (P 6, L 157–160), and uncertainties due to vegetation cover on the palsas (P 7, L 172–176):

"*Change analysis based on UAS DSMs is sensitive to small changes in vegetation structure and differences in shadows/lightning conditions. No notable changes in the vegetation cover of the palsas were observed during the annual visits to the sites by the authors. However, it is possible that small differences in vegetation cover because of, for example, trampling or vegetation growth may have affected the height and volume changes derived from the DSMs.*"

We also added Table S3 with details including acquisition dates, spatial resolutions, number of GCPs, and XYZ errors in the Supplement.

Unfortunately, we do not yet have fixed GCPs, which would allow a more accurate comparison of the different UAS DSMs (as suggested by Fraser et al., 2022). Thus, the only objects that could potentially be used as "stable" reference points at our sites were boulders around the mire areas. The issue with them is that they are located at the edges of the areas covered by the UAS surveys, where the "edge effect" causes uncertainties in the elevation. We added this point on P 7, L 177–178:

"*Due to the lack of stable surfaces unaffected by the "edge-effect" in the UAS data, we were unable to correct for potential global offsets between the DSMs*."

48. 162 ff: Please explain in more detail why SnowModel is a suitable tool to reproduce snow dynamics in the extremely challenging environment on top of a palsa, i.e. present some key elements of the model physics, in particular on wind redistribution. The validation provided in favor of the model relies on an unpublished master thesis which I wasn't able to access with a quick Google search. Please provide more details on this work in the manuscript, i.e. include the main findings of the thesis in this study. From the information provided, it is not possible to assess whether the modeled snow data allow for a sounds assessment of long-term trends, thus also affecting the Results section. Additional validation on snow onset and melt-out could possibly be obtained from remote sensing data, e.g. Sentinel-2, at least for years with infrequent cloud cover in the respective periods.

In addition to the validation presented in the Master's thesis of A. Störmer (2020), we are confident that the SnowModel (Liston and Elder, 2006) is suitable (although not perfect) tool to estimate snow dynamics on palsas because it takes into account different meteorological variables and climatic processes, local topography and vegetation height.

We agree that the description of the SnowModel was too short in the first submitted version of the manuscript. We did not describe it in more detail, as it was a complementary addition to the analysis to check whether the local snow depth values would show better correlation with the ALT compared to the values measured at the meteorological station in Kilpisjärvi. We also agree that the unavailability of the thesis, which is referred to, makes it impossible for the reader to check further details. Thank you for pointing this out! To address this issue, we included in supplementary materials a more detailed description of the SnowModel, the parameters used, potential issues, and a comparison of our SnowModel results to the measured values, which A. Störmer used in his thesis. We did not include the key elements of the model physics as they are thoroughly explained in the article by Liston and Elder (2006), which is freely available online.

Validation of the snow-on and snow-off dates with optical satellite data, such as Sentinel-2 or even Landsat, would be a good addition. Unfortunately, frequent cloud cover over the study areas resulted in very few images that could be compared with the dates derived from the snow depth data at the Kilpisjärvi Weather Station. The few available images for the snow-off dates confirmed what could be also expected based on the lower accumulation of snow on the palsa surface and game camera recordings at Peera from autumn 2016 until end of summer 2017, that actual snow-off dates are generally earlier for palsas than observed at the weather station. Fewer images are available for the snow-on dates at our sites, but based on the game camera images, the difference is much less pronounced. We included this point to the description of snow parameters on P 8, L 201–205:

"Snow accumulation *is likely lower, and snow melts earlier* on wind-exposed palsas than at the *weather* station. *As we were not able to validate snow onset and melt dates from freely available optical satellite data, due to frequent cloud cover, we assumed that the differences in the timing of snow cover between the station and the palsas are similar every year. However, to account for thinner snow cover on the palsas,* we estimated the local snow depths at our sites using SnowModel (Liston and Elder, 2006)."

49. Sect. 4.1 The negative trend for the second site is very interesting – please add 1-2 sentences to highlight the procedure again, in particular that only values from the TOParea, i.e. the still stable part in later years, are compared. It is easy to miss this as a casual reader.

We added a subscript "$_{TOP}$" whenever we specifically refer to the top-of-palsa (TOP) ALT values.

We also added more information about our motivation to focus on the TOP-areas (P 6, L 133–135) and included analyses using all ALT values ≤ 1m as described in the earlier reply to the Reviewer 1 comment #9.

50. Fig. 4: It is not really clear to which site the regression parameters and the R2 values belong, the one on the top also has a different color in some of the plots?

We agree that the colours of the texts within Figure 4 were confusing. We edited the colours of the equations and $R^2$ values so that the ones for Peera will be in black, and the ones for Laassaniemi are in light blue similar to the point symbols.

51. Sect. 4.2 I don't think it makes sense to present correlations that are not statistically significant, even if there is a trend. This is exactly the point of a statistical significance test. So for me the main conclusion of this section is that ALT is not strongly controlled by any of the tested parameters, except for the ones pointed out by the authors as significant. But also for these, it would be good to discuss the level of significance some more. This in itself is a very important result, in particular that the clear decrease in ALT for the second site does not seem to be controlled by larger-scale climatic drivers, but more by local factors which the authors cannot quantify at this point. I do not question the analysis (except perhaps the snow data, see above), but I think this section needs to be rewritten to some extent.

We kindly disagree about the relevance of including non-significant correlations, such as relationships between the ALT and air temperatures (Fig. 4 a, b, d, and e) in this case. The air temperatures have been found to be important factors controlling the ALT in palsas in other studies (Åkerman and Johansson, 2008; Sannel et al., 2016; Mamet et al., 2017), which makes it interesting that the relationships with these parameters were not that strong at our sites. And that is why we think it is important to present them in the results.

We revised Section 4.2 so that it states more clearly, which correlations were statistically significant, and included $R^2$ values of the variables mentioned in the text, as advised by Reviewer 1.

52. Table 2: can you add the corresponding data, i.e. 2016 and 2021, from the dGPS surveys to this table! The difference in absolute values seems to be significant between the two methods, so having a direct comparison of the same time slices is important.

The UAS DSMs, which were used to derive the values presented in Table 2, cover larger areas than the RTK-GNSS surveys from the same years. For this reason, we did not include the RTK-GNSS DTM values for 2016 and 2021 in the same table. Instead, we decided to add a new Section 4.5 (P 18–19), where we compare palsa height changes 2016–2021 based on the two methods. In this new section, we also included the RMSEs for the elevation and height values between the RTK-GNSS and UAS DSMs.

53. Fig. 6: why are there two color legends (one in meter, the other in cm)? I think it could also be good to adjust the color scales and not use confining max-min-values. Right now one mainly sees the areas of full collapse, but it is equally important to be able to assess to what extent the main areas of the palsa have subsided. Furthermore, the authors should clarify to what extent the increases near the palsa edge are due to vegetation (i.e. is there vegetation of such height at all? Is the first survey taken after leaf-fall and the second before?), or the result of consistency issues between the DEM's, like global shifts, tilts or rotations (see also comment on DEM accuracy above).

We wanted the same colours to indicate the same values for both palsas. To achieve this, we used 25 classes with 10 cm intervals (except for the min and max classes). The continuous legend for the changes in metres is a compromise, as showing all the 25 classes in the figure legend does not seem reasonable to us, and using fewer classes hinders changes at Laassaniemi. We acknowledge that it is difficult to assess height changes in the core areas of the palsas, however.

The leaves were still on during the UAS surveys in 2016 and 2021. The vegetation height varies from zero centimetres to ca. 50 cm. The shrub cover is particularly high at the northern edge of the Peera palsa. Although some displacement of the vegetation may have occurred, the apparent increases at Peera are more likely due to differences in the quality of the UAS data and the resulting DSMs. At the northern edge of the Laassaniemi palsa, the change was indeed due to vegetation growth, as the common cotton grass has expanded rapidly within the thermokarst pond.

As mentioned in the reply to Reviewer 1 comment #26, we have rearranged Figure 6 and enlarged legend texts. In addition, we added contour lines to Figures 6 a and b showing the absolute height changes, so that it is easier to assess the changes in the core areas of the palsas. We also added two photos to illustrate what kind of vegetation grows in the areas of apparent height increase. In the text, the effects of vegetation are clarified on P 16, L 304–308 as follows:

"*The apparent increase in the height at the northern edge of the Peera palsa is likely a result of differences in how shrubs (Fig. 6c) and their shadows influenced the point cloud generation from the RGB-data, and consequently the DSMs. At the northern edge of the Laassaniemi palsa, the increase in height was caused by the rapid expansion of common cotton gras within the adjacent thermokarst pond (Fig. 6f).*"

54. 259ff: I am not sure about these correlations, is there any statistical significance? Also, the authors write that a higher value of snow onset (=later snowfall) correlates with a higher degradation rate for the second site (true?), but there is no correlation to e.g. fall air temperature? A later snow onset should rather lead to more ground cooling, except when the air temperatures are above freezing. If the data are like that, it is important to state this result, but the authors should check to what extent such correlations are statistically supported.

We rephrased the paragraph about the correlations between the annual area loss rates and the climatic parameters, so that it states more clearly the lack of significant correlations and that the regression results are rather indicative because of the low number of samples. The $R^2$ values of the correlations mentioned in the text are now also included. In addition, we moved Table A2 into the main text as mentioned in the replies to Reviewer 1 comments #29 and #31.

Indeed, the regression results in Table 6 (old Table A2) imply that at Peera, the area loss rates are higher in periods with later snow onset, although the correlation is not significant. At Laassaniemi, this relationship is opposite and is in line with the negative correlation between the ALT and snow

onset at both sites (thinner ALT after later snow onset in the preceding winter). The opposing relationships are in agreement with our interpretation of the results concluding that the palsa area loss at Peera is more related to the changes in air temperatures, and at Laassaniemi, the area loss is more related to the changes in snow cover.

We did not discuss the correlation with autumn air temperatures. However, quick analysis showed that the area loss rates did not correlate with autumn (Sept., Oct., Nov.) air temperatures at either site. The analysis showed also that the mean autumn air temperatures have increased significantly 1960–2021 (+ 0.3 C per decade, $R^2 = 0.1$, $p = 0.012$), and the snow cover onset correlates with autumn temperatures ($R^2 = 0.35$, $p < 0.001$).

The revised paragraph on P 22: L 382–389 reads as follows:

*"The full list of $R^2$ and p-values of the correlations between the annual area loss rates and climatic parameters is presented in Table 6. Because of the low number of samples (only four periods), the results of the linear regression analyses between the annual area loss rates and climatic parameters are only indicative. None of the correlations were statistically significant at the 95 % confidence level (Table 6). At Peera, three parameters related to the winter air temperatures had the highest $R^2$ and p-values < 0.1. These parameters were MWAT ($R^2 = 0.88$), √FDD ($R^2 = -0.88$), and the number of days with air temperature < -10 °C ($R^2 = -0.87$). At Laassaniemi, the area loss rates had very little correlation with the climatic parameters; the lowest p-values were only around 0.3 for the snow cover onset ($R^2 = -44$) and snow cover duration ($R^2 = 0.46$)."*

**References**

Fraser, R., Leblanc, S., Prevost, C., and van der Sluijs, J.: Towards Precise Drone-based Measurement of Elevation Change in Permafrost Terrain Experiencing Thaw and Thermokarst, Drone Syst. Appl., https://doi.org/10.1139/dsa-2022-0036, 2022.

Liston, G.E. and Elder, K. A.: Distributed Snow-Evolution Modeling System (SnowModel), J. Hydrometeorol., 7, 1259–1276, https://doi.org/10.1175/JHM548.1, 2006.

Mamet, S., Chun, K.P., Kershaw, G.G.L., Loranty, M.M., and Kershaw, P.: Recent Increases in Permafrost Thaw Rates and Areal Loss of Palsas in the Western Northwest Territories, Canada, Permafrost Periglac., 28, 619–633, https://doi.org/10.1002/ppp.1951, 2017.

Sannel, A.B.K., Hugelius, G., Jansson, P., and Kuhry, P.: Permafrost Warming in Subarctic Peatland – Which Meteorological Controls are Most Important? Permafrost Periglac., 27, 177–188, https://doi.org/10.1002/ppp.1862, 2016

Störmer, A.: Modelling snow distribution over discontinuous permafrost related to climate change in Kilpisjärvi, Finnish-Lapland, M.S. thesis, Faculty of Natural Sciences, Gottfried Wilhelm Leibniz University of Hannover, Germany, 2020.

Tomhave, L.: Palsa Development and Associated Vegetation in Northern Finland. B.S. thesis, Faculty of Natural Sciences, Gottfried Wilhelm Leibniz University of Hannover, Germany, 2018.

Åkerman, H.J. and Johansson, M.: Thawing Permafrost and Thicker Active Layers in Sub-arctic Sweden, Permafrost Periglac., 19, 279–292, https://doi.org/10.1002/ppp.626, 2008.